

# Hydrological modelling on atmospheric grids; using graphs of sub-grid elements to transport energy and water

Jan Polcher[1], Anthony Schrapffer[2,1], Eliott Dupont[3], Lucia Rinchiuso[4,5], Xudong Zhou[6], Olivier Boucher[3], Emmanuel Mouche[4], Catherine Ottlé[4], and Jérôme Servonnat[4]

[1]LMD-IPSL, CNRS, Ecole Polytechnique, Route de Saclay, 91128 Palaiseau Cedex, France
[2]Centro de Investigaciones del Mar y la Atmósfera (CIMA), CONICET-UBA, CNRS, Intendente Güiraldes 2160 - Ciudad Universitaria - Pabellón II - 2do. piso (C1428EGA) C. A. Buenos Aires, Argentina
[3]Institut Pierre-Simon Laplace (IPSL), CNRS, U. Sorbonne, Paris, France
[4]LSCE-IPSL, CEA, CNRS, U. Paris-Saclay, Gif-sur-Yvette, France
[5]METIS-IPSL, CNRS, U. Sorbonne, Paris, France
[6]Institute of Industrial Science, University of Tokyo, Tokyo, Japan

**Correspondence:** Jan Polcher (jan.polcher@lmd.ipsl.fr)

**Abstract.**

Land Surface Models (LSMs) use the atmospheric grid as their basic spatial decomposition because their main objective is to provide the lower boundary conditions to the atmosphere. Lateral water flows at the surface on the other hand require a much higher spatial discretization as they are closely linked to topographic details. We propose here a methodology to automatically

tile the atmospheric grid into hydrological coherent units which are connected through a graph. As water is transported on sub-grids of the LSM, land variables can easily be transferred to the routing network and advected if needed. This is demonstrated here for temperature. The quality of the river networks generated, as represented by the connected hydrological transfer units, are compared to the original data in order to quantify the degradation introduced by the discretization method. The conditions the sub-grid elements impose on the time step of the water transport scheme are evaluated and a methodology is proposed to

find an optimal value. Finally the scheme is applied in an off-line version of the ORCHIDEE LSM over Europe to show that realistic river discharge and temperatures are predicted over the major catchments of the region. The simulated solutions are largely independent of the atmospheric grid used thanks to the proposed sub-grid approach.

## 1 Introduction

Lateral water transport over continents plays an important role in the Earth system but its implementation in models focusses

on different objectives depending on resolution. In global Earth System Models (ESM), tailored to address climate change issues, the main need is to transport the excess water over land to the oceans so as to close the water cycle. Because of their coarse resolution the main focus will be on the largest rivers. Regional ESM, as those used for process studies and downscaling of climate projections, will usually attempt to reproduce more details in the continental water cycle. Lateral water transports will thus also serve to predict levels of in-land water bodies or inundations and the impact of freshwater flows on coastal

processes. Finally for km-scale ESM currently being developed to better represent rainfall and convective processes, lateral




flows allow to redistribute moisture along hill-slopes (Fan et al., 2019). At these resolutions the hypothesis that evaporation is only fed by local precipitation is not valid any more. Thus rain falling on mountain slopes needs to flow into the valleys where the vegetation is located which will be able to evaporate it. These are resolutions where hill-slope processes will start to be important.

Furthermore rivers also transport energy and biogeochemical species (Liu et al., 2020; Lauerwald et al., 2017). Thus their correct representation allows to close the associated global cycles of the Earth system and improve the coupling between continental and oceanic processes. To enhance the representation of coastal processes in ESMs at all scales, the energy and nutrient contribution by rivers will be a major topic in the years to come.

  Land surface models, the components within ESMs dealing with continental processes, have implemented over the last 30

years a very uni-dimensional vision of the water and energy processes at the surface. The main driver of their development was to provide the lower boundary to the atmosphere. As a consequence they have also adopted the spatial discretization of the atmosphere so as not to introduce any discontinuity in this important coupling. The lateral water transport cuts across this one dimensional vision and also challenges the use of the atmospheric grid. Indeed, lateral water movements require often higher resolution than the atmosphere as topographic features are a stronger constraint for the flow of water on land than for

the atmosphere. The hydrological community have been free of this constraint of the coupling to the atmosphere and could adopt appropriate spatial discretization, which is often kilometric, for the representation of rivers.

  ESMs have adopted over the last decades two different and complementary approaches to try and deal with the lateral water transport on continents. The first and most widespread approach is to abandon the atmospheric grid and interpolate the fields of water exiting the one dimensional soil moisture (generally surface runoff and deep drainage) towards the grid

which will be used to simulate river flows (Decharme et al., 2012; Branstetter, 2003). From this grid the discharge to the seas is then transmitted to the ocean model. The river routing model becomes an independent component of the ESM meaning it can be developed and validated separately. Sharing this component with other groups is also facilitated (Kauffeldt et al., 2016). The main drawback of this approach is that it creates a division in the interactions between the vertical and horizontal motions of water on continents. It complicates the representation of such important processes as floodplains or irrigation. The

interpolations between the atmospheric and routing grids will need to fulfil the conservation principles and preserve gradients in surface processes.

  The second approach is to use directly the atmospheric grid for the lateral flows (Miguez-Macho et al., 2007). This has been successfully applied in regional and km-scale ESMs for which the horizontal atmospheric grid are compatible with hydro-logical processes. The methodology has been particularly successful for flood forecasting where lead time for predictions are

short (Yucel et al., 2015). A complementary methodology is to use a hydrological tiling of the atmospheric grid to achieve the effective resolution needed (Ngo-Duc et al., 2007; Clark et al., 2015). The general principle is to keep the vertical movements of water on the atmospheric grid but distribute the excess moisture over hydrologically consistent and connected tiles so that it can flow horizontally. In principle this methodology should be able to bridge the gap between coarser atmospheric resolutions and the level of detail needed for surface flows while keeping a close link between the vertical and horizontal processes.

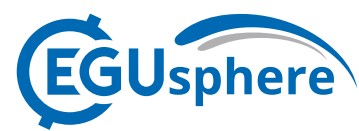

55 Using the nomenclature proposed by Yamazaki et al. (2013) this last method can be labelled hybrid river models. It uses the vector-based methodology within the atmospheric grid as water is transported between unit-catchments and then uses a grid-based approach when the flow leaves the mesh. The combination of both yields graphs of hydrological transfer units (HTU) along which the water will flow within and from grid to grid. In the list of criteria established by Kauffeldt et al. (2016) to classify large-scale hydrological models, a hybrid routing addresses in particular the two linked to the grid. It should be

60 able to deal with any atmospheric grid and in particular those based on the more complex icosahedron (Dubos et al., 2015) or cubed-sphere (Kim et al., 2021). The flexibility in resolution is given as the hydrological information which cannot be resolved with the grid-based flow is treated with the vector-based transport.

 Preserving the link between the atmospheric resolution and the sub-grid hydrology facilitates the representation of all the processes which involve exchanges between vertical and lateral water movements. It allows to represent floodplains and their

65 impact on evaporation (Companion paper Schrapffer et al. 2022) or water extractions for human activities (de Rosnay et al., 2003; Zhou et al., 2021). In this context the stream temperature can also be more easily simulated as the energy balance performed on the atmospheric grid can directly interact with open water areas which are part of the lateral flows.

 The proposed study explores the numerical properties of such a hybrid routing scheme and in particular how it handles different atmospheric grids. In a first step we will show how the graph of HTU can be built and which properties of the

70 hydrological network need to be preserved. Then we will present criteria which allow to verify the fidelity of the HTU graph and in particular how many sub-grid elements are needed for different resolutions of the atmospheric grid to preserve the original hydrological information. Once water is transported, criteria are needed to select a time step which ensures that the numerical solution converges. Finally we will show with ORCHIDEE that the simplification of the digital elevation model introduced by the transport on a graph of HTUs is small compared to the uncertainty in the atmospheric forcing. The methodology will also

75 be used to demonstrate the value of a simple implementation of stream temperature.

## 2 The river flow model

Before presenting the construction of the graph of hydrological transfer units (HTU), the equations used to transport water and heat along the network of rivers are presented. They give indications on the properties of the graph which need to be preserved.

### 2.1 Water transport

80 The flow of water occurs on a directional graph towards the outflow point (Diestel, 2012). As for the moment the focus is on simulating river flows, it is the $stream$ reservoirs which connect the various HTUs in the graph. Each HTU $i$ is contained in only one atmospheric grid box $\widehat{i}$ and covers the fraction $\delta_{i,\widehat{i}}$ with the sum of all HTU areas equal to the continental surface provided by the atmospheric model. Thus for the fraction of land in the grid we have : $\sum_{i \in \widehat{i}} \delta_{i,\widehat{i}} = 1$. This means that the HTUs are a connected supermesh of the land fraction within the atmospheric grid (Farrell et al., 2009) and which represent the river

85 graph.

 As the graph is directional we can pose the following indexing convention :




- $i+1$ is the downstream store of HTU $i$. Because we are in a directional graph $i+1$ is unique and at one point should be the ocean or a water body for endorheic basins.

- For an HTU $i$ there is an ensemble of upstream HTUs which will be denoted with $\{i-1\}$. For a basin head the ensemble will be empty.

- Fluxes connecting two HTUs are denoted with half indices. The flux leaving HTU $i$ is placed at $i+1/2$

Given the above notation, the prognostic equations for water transports are given by :

$$\frac{\partial W_{i,stream}}{\partial t} = \sum_{j \in \{i-1\}} (Q_{j+\frac{1}{2},slow} + Q_{j+\frac{1}{2},fast} + Q_{j+\frac{1}{2},stream}) - Q_{i+\frac{1}{2},stream} \tag{1}$$

$$\frac{\partial W_{i,fast}}{\partial t} = \delta_{i,\widehat{i}} R_{\widehat{i}} - Q_{i+\frac{1}{2},fast} \tag{2}$$

$$\frac{\partial W_{i,slow}}{\partial t} = \delta_{i,\widehat{i}} D_{\widehat{i}} - Q_{i+\frac{1}{2},slow} \tag{3}$$

$$\tag{4}$$

Here $R_{\widehat{i}}$ is the surface runoff (i.e. water which does not infiltrate into the unsaturated zone of the LSM) on the atmospheric grid and $D_{\widehat{i}}$ the drainage exiting the soil moisture reservoir. Both of these fluxes are computed by the land surface model and thus behave according to the assumptions made there. The fluxes out of the three reservoirs are $Q_{i+\frac{1}{2},slow}, Q_{i+\frac{1}{2},fast}, Q_{i+\frac{1}{2},stream})$ and are expressed in $kg/s$. They are defined as follows :

$$Q_{i+\frac{1}{2},X}^{t} = \frac{W_{i,X}^{t}}{\lambda_{i,X} \, g_X} \text{ with } X \in \{stream, fast, slow\} \tag{5}$$

The flow is thus given by the reservoir's water mass divided by the residence time. This characteristic time of each reservoir is the product of a geometric property, the topographic index symbolized by $\lambda$ in $km$ , and a constant in $s/km$ which needs to be determined. We chose to distinguish the topographic index of the stream flow ($\lambda_{i,stream}$) and the one for the aquifers of the HTU ($\lambda_i$). The first one can be evaluated based on the geometry of the main river within the HTU. The other one provides a more aggregated view of the geometry of the HTU. The general principle for computing these parameters is :

$$\lambda = \sqrt{\frac{d^3}{dz}} \tag{6}$$

where $d$ is the relevant length over which the water flows within the HTU and $dz$ the elevation change along that path. In the section covering the construction of the HTU graph the computation of the topographic indices is detailed for each case (see section 3.5).

## 2.2 Stream temperature

The advection of heat content with lateral water transport is given by the advection of the aquifer temperatures and the interactions with the surrounding landscape and the atmosphere. In order to implement these processes, an initial temperature has to





be chosen for the water in the fast and slow reservoirs. The assumption here is that they are determined by the temperature of

115 the soils of the corresponding atmospheric grid. Our initial assumption is :

$$T_{i,fast} = T_{\widehat{i},up} \tag{7}$$

$$T_{i,slow} = T_{\widehat{i},low} \tag{8}$$

where $T_{\widehat{i},up}$ is the soil temperature averaged over the top $0.3\,m$ and $T_{\widehat{i},low}$ the mean value over the $3.5-17.5\,m$ layers. Currently

the deepest node in the soil temperature diffusion model in ORCHIDEE is $17.5\,m$ thus imposing a lower limit on the depth for

the temperature of the slowest aquifer. This situation can evolve in future versions of ORCHIDEE.

Evaluating the stream temperature through the advection of heat has to deal with the singularity arising when the reservoir

content goes to zero. A relaxation towards the upper soil temperature was chosen to deal with this indetermination. This allows

to write the following set of equations :

$$\frac{\partial T_{i,stream}}{\partial t} = \frac{1}{W_{i,stream}} \sum_{j \in \{i-1\}} (Q_{j+\frac{1}{2},slow}T_{j,slow} + Q_{j+\frac{1}{2},fast}T_{j,fast} + Q_{j+\frac{1}{2},stream}T_{j,stream})$$
$$- \frac{1}{W_{i,stream}}Q_{i+\frac{1}{2},stream}T_{i,stream} + K(T_{\widehat{i},up} - T_{i,stream}) \tag{9}$$

and

$$K_i = \frac{1}{1 + a\rho^{-1}W_{i,stream}/A_i} \tag{10}$$

where $A_i$ is the area of the HTU and is used to transform the water mass in the stream multiplied by its density ($\rho$) into an

equivalent height. $a$ is a scaling parameter which allows to adjust the role of the relaxation term and thus explore the relative

importance of initial temperatures and interactions with the environment surrounding the river in the determination of $T_{i,stream}$.

If $a$ is large, Eq 9 will behave like a pure advection scheme and only adjust the temperature of stream with shallow water levels

to the top soil temperatures. As $a$ decreases the interaction of the stream temperature with its environment will increase. This

relaxation will mostly be active on HTUs with low stream reservoir which is physically reasonable. In areas where water flows

in small streams the temperature of the water will be close to the one of the upper soils. This simple relaxation obviously does

not cover the complex interactions of lakes or large rivers with the atmosphere and which are known to impact strongly stream

temperature.

The numerical solution of these equations is discussed later (Section 5) as it depends on the definition of the HTUs.

## 3    Building the graph of Hydrological Transfer Units (HTU)

This section presents the methodology for constructing Hydrological Transfer Units (HTUs) and connecting them to build a

hydrological network suitable to simulate surface water transport. In contrast to the tiling of the atmospheric grid for vegetation

(Ducoudré et al., 1993; Koster and Suarez, 1992), the connectivity of the sub-grid elements needs to be preserved and form a

convergent hydrological graph. It is an evolution of the method discussed by Nguyen-Quang et al. (2018).

 

**Table 1.** The atmospheric grids and land/sea masks for which HTU graphs were built to test the methodology.

| grid name | Source | Resolution | Projection |
|-----------|--------|-----------|-----------|
| WFDEI | WFDEI forcing dataset Weedon et al. (2014) | $0.5°$ | Regular longitude and latitude |
| E2OFD | MSWEP precipitation product (Beck et al., 2017) | $0.25°$ | Regular longitude and latitude |
| MEDCORDEX | MED-CORDEX configuration | $20km$ | Lambert conformal |
| EuroCORDEX | EuroCORDEX configuration | $11km$ | Cassini projection |

For the present study, the methodology for tiling the atmospheric mesh into graphs of HTUs is exemplified on 4 grids covering the Euro-Mediterranean region (Table 1) which are either used for off-line or coupled simulations of ORCHIDEE. The domain of computation used here is at least from 20° East to 60° West and 20° to 60° North. To illustrate the diversity of grids a small sample over the lower Seine is provided in Figure 1. Only rectangular meshes are considered here but the methodology can be extended to triangular ones. The companion paper (Schrapffer et al. 2022), which presents the floodplains parametrization, uses HTU graphs built for two atmospheric grids over South America.

The methodology works on any atmospheric grid as long as the polygons constitutive of the mesh are provided together with the land/sea mask.

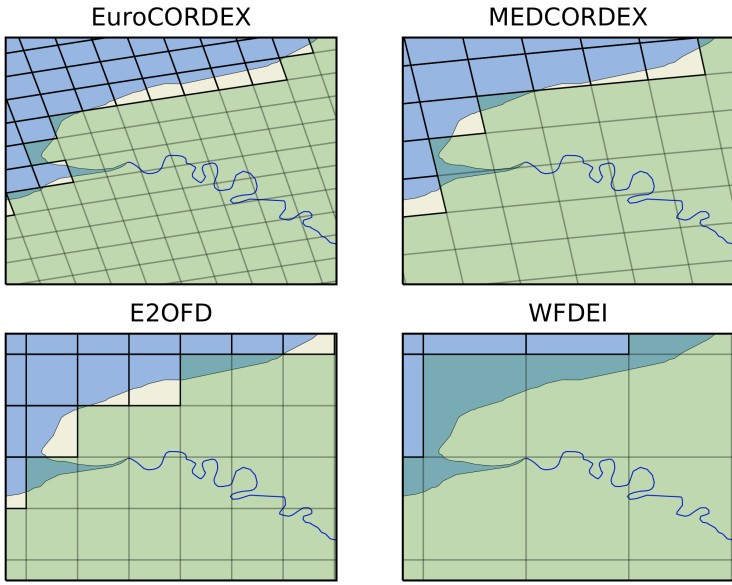

**Figure 1.** Samples over the lower Seine basin of the four atmospheric grids considered here (Table 1). The green colour indicate which meshes are land for the atmospheric model overlaid over the actual land outline in yellow.

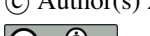



**Table 2.** The hydrological data sets used in this study to evaluate the building of routing graph and the simulated river discharge

| Data set name | Source | Resolution | Domain covered | Constants used [$s/km$] | | |
|---|---|---|---|---|---|---|
| | | | | $g_{stream}$ | $g_{fast}$ | $g_{slow}$ |
| FV | Fekete et al. (2000) | $0.5°$ | Global including Antarctica | 6.0 | 80.0 | 600 |
| HydroSHEDS | Lehner and Grill (2013) | $30 acrsec$ | South of $60° N$ and excluding Antarctica | 6.3 | 80.0 | 600 |
| MERIT | Yamazaki et al. (2019) | $60 arcsec$ | Globe without Antarctica | 6.3 | 80.0 | 600 |

## 3.1 Hydrological digital elevation models

To perform the HTU decomposition and compute their properties a hydrological digital elevation model (HDEM) is needed. The minimal information required is elevation, flow direction, flow accumulation and distance to the ocean for each pixel. The elevation should be hydrologically consistent in the sense that no flow should lead water to gain elevation. As we will show below, this is not a strict requirement. The three data sets listed in Table 2 fulfil these criteria and have been used here to test 155 the methodology. They needed to be standardized as each of these HDEM had a different set of variables required and were not using the same conventions.

The original HDEM used in IPSL's ESM is from Fekete et al. (2000) which is at $0.5°$ resolution. It has been enhanced by flow directions in Antarctica in order to carry meltwater to the ocean and close the global water cycle. Its low resolution was perfectly suited for the global models used in climate studies such as Ngo-Duc et al. (2007). The advent of higher resolution 160 atmospheric models means that its usefulness is diminishing. For regional applications the HydroSHEDS data (Lehner and Grill, 2013) was used in ORCHIDEE over the Mediterranean region Nguyen-Quang et al. (2018). The fact that this HDEM is not global means that it is only applicable for regional studies which do not cover the arctic region. In order to bridge this gap between global coarse and regional high resolution HDEMs we have chosen to also use a $60 arcmin$ version of MERIT (Yamazaki et al., 2019). This version of MERIT was obtained by upscaling from the high-resolution flow direction data (3 165 arcsec MERIT Hydro) with the Iterative Hydrography Upscaling (IHU) method (Eilander et al., 2021).

## 3.2 Supermesh between atmospheric grids and HDEM

The HTUs are built from the supermesh (Farrell et al., 2009) between the atmospheric grid and the hydrological digital elevation model (HDEM). The initial step is to create this supermesh by calculating the set of polygons of the HDEM grid overlapping with the coarser mesh. The result is for each atmospheric grid point the list of polygons of intersecting polygones and their 170 area. At the borders of the atmospheric grid a number of small polygons with their areas and flow direction will be created. This generates some numerical noise but which will be absorbed during the HTU construction process. More importantly it allows to preserve catchment areas and the diversity of flow directions out of the atmospheric grid.





This process is performed using the *SphericalGeometry* library (https://github.com/spacetelescope/spherical_geometry) implemented in Python, which allows to calculate intersections of polygons over a sphere. This supermesh between the atmo-

spheric grid and the HDEM can be saved into a NetCDF file to avoid having to perform this calculation again on the same combination of grids. The advantage of this procedure is that it is applicable to any atmospheric grid even unstructured ones. This is particularly important as atmospheric models evolve towards more complex grids (Dubos et al., 2015).

### 3.3  First construction of HTUs

A first set of coarse HTUs is build by joining all polygons of the supermesh which flow out of the atmospheric grid at the

same point. Their upstream area is computed according to the HDEM. This provides a first set of HTUs which is quite coarse but ensures that all rivers and flow directions out of the atmospheric grid are preserved. The example in Figure 2c) ( for $nbmax = 18$) still carries some elements of this decomposition with the HTU 18 flowing out of the Eastern part of the Northern edges of the atmospheric grid. These small HTUs will not transport large water amounts but ensure that the total upstream area of rivers to which they contribute remains correct. At this stage all pixels which contribute to the main outlet of the Rhone in

the lower left corner are still in a single and same HTU. This first step will conclude for the case illustrated in Figure 2 with as many HTUs as there are arrows pointing out of the grid.

### 3.4  Sub-division of HTUs

An algorithm is needed to sub-divide the larger HTUs to better represent the river network within the atmospheric mesh. The objective is to divide the HTU at important confluences. There are two types of confluences which need to be considered:

– Two large rivers (large global upstream area) meet as illustrate in Figure 2a) around Valence. Figure 2b) shows three rivers with a large global flow accumulation joining the Rhone river in the grid.

   – A local river (large local upstream) joins a large river. This is for instance the case for HTU 8 in Figure 2c) which corresponds to La Barberolle flowing into the Rhone river in Valence.

The section is made at the confluence pixel between the main river and its tributary. This produces 3 new HTUs : (1) the

part of the main river upstream of the confluence and with this pixel of the HDEM as it outflow location, (2) the downstream part of the main river which keeps the initial outflow pixel and (3) the tributary which has as outflow pixel the one before the confluence. If the subdivision (1) or (2) are too small, the HTU is only divided into two parts : the HTU of the main river and the confluence. The tributary will now flow directly to the outflow point of the original HTU and thus potentially creating a small topological error. The aim is to avoid generating too small HTUs on the main rivers.

This algorithm is iterated until one of two conditions is met :

   – the tributary has the 4th highest global flow accumulation.

   – The local upstream area is less than 10% of the grid area.



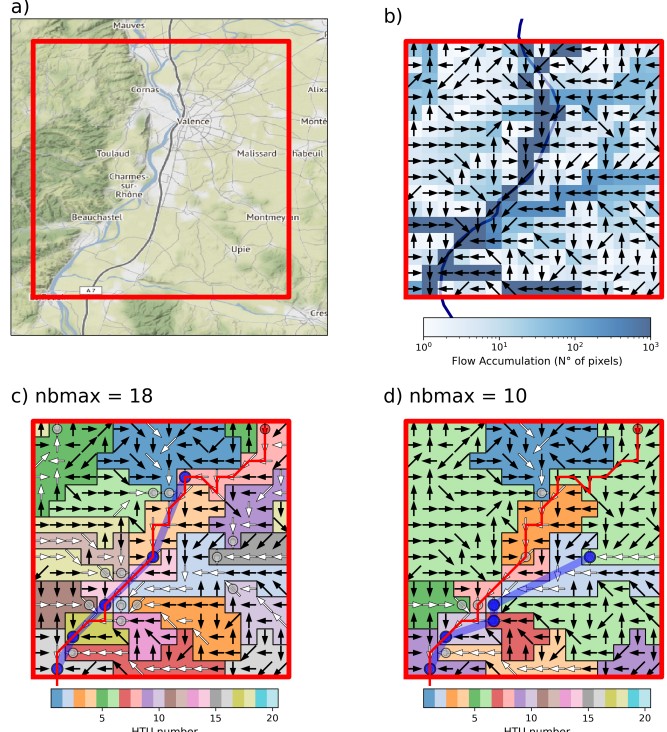

**Figure 2.** A sample case of a regular atmospheric grid decomposed into HTUS over the Rhone valley. a) The geographical context of the Rhone at Valence, France. b) Flow directions and accumulation for the HDEM pixels overlapping the atmospheric grid. c) Resulting HTU decomposition using $nbmax = 18$. d) As previously but for $nbmax = 10$. The white arrows within each HTU highlight the river segment used to determine the geometric properties used to compute the residence time of the stream reservoir. The grey dots indicate the location where one HTU flows into the downstream one. In the two lower panels the red line highlights a sample river segment in the HDEM as is used to statistically evaluate the river graphs (Section 4). The blue line is the corresponding river segment but now in HTU space.

## 3.5 Computing topographic indices

The residence time of water in the stream reservoirs is given by the topographic index ($\lambda_{i,stream}$) and a constant ($g_{stream}$) as shown in equation 5. $\lambda_{i,stream}$ is determined using equation 6 with the length and elevation change of the river between the inflow pixel with the largest upstream area and the outflow of the HTU. If the HTU is a headwater, then the longest path is used until an elevation change larger than 30% on the HDEM is reached when moving upstream. This avoids taking into account the steepest parts of the catchment. The aim is to capture the geometrical feature of the main river flowing through the unit. In Figure 2d) this would be the elevation change and length along the pixels with a white arrow within each HTU.

For the fast and slow reservoirs, the topographic index $\lambda_i$ should represent the general characteristics of the whole HTU and its complexity when computing the outflow in equation 5. Therefore, another approach is used based on the hypothesis that



there is a set of small streams or subterranean flows contributing to the main stream of the HTU. Starting from the topographic index computed on each pixel of the hydrological grid, sums are computed along all streams up to the outflow point of the HTU. These sums are then averaged to provide an integrated property for the HTU. This allows to produce a $\lambda_i$ which is representative of the soils and hill-slopes surrounding the river. This was found to work best in the proposed set-up but is only a crude simplification of what is known of hill-slope and riparian processes.

If other variables of the LSM, soil moisture for instance, are also simulated at the HTU level then the hill-slope processes which govern the riparian water exchanges could be parametrized. As suggested by Picourlat et al. (2022) the geometric characteristics of the HTU and its soil moisture can be used to predict the flow of surface groundwater into the river.

## 3.6 Reaggregation

At this stage, the atmospheric grid can contain more HTUs than requested by the user with the $nbmax$ parameter. The user may want to choose a simpler river network in order to reduce the memory footprint of the routing scheme. This depends on the configuration best suited for the user's need. But we would not recommend to select a values of $nbmax$ below 8 on a rectangular grid as then the number of outflow directions of the meshes is limited. The reaggregation step of the routing pre-processor is to reduce the number of HTU on all grid points of the atmospheric to the $nbmax$ value chosen by the user. The merger of HTUs which will be performed by always favouring the largest HTUs or those with the largest upstream area and try to preserve the diversity of outflow directions out of the atmospheric grid. The elimination of the smallest HTUs is performed in 5 steps until the $nbmax$ value is reached :

1. Merge all HTUs of an atmospheric grid which flows into the ocean,

2. Merge HTUs which flow to the same neighbouring grid by starting with the smallest. This reduces the border noise by merging the smallest HTUs which have been generated by the supermesh methodology.

3. Merge HTUs which belong to the same river and flow out of the mesh. As this is also performed for HTUs flowing out in different directions it generates topological errors.

4. Merge HTUs which flows into a downstream HTU within the same atmospheric grid,

5. Finally a brute-force method is used to merge the smaller HTUs until $nbmax$ is reached. If this method has to be applied to HTUs with an area higher than 5% of the grid box the user is warned that a higher $nbmax$ should be considered.

During the reaggregation step, the HTUs do not need to be connex when merged. This leads to situation like HTU 6 in Figures 2d) which is composed of areas East and West of the Rhone.

As highlighted in the discussion above, in order to simulate a realistic river discharge the length and slope of the main rivers need to be well preserved in the HTU decomposition. It is clear that if the truncation steps have to be carried too far because of a poor choice by the user, a poor quality graph will be obtained and a reliable simulation of the stream flow cannot be expected. Below we will present a methodology which allows to verify the quality of the graph and estimate an optimal number of HTU for a given atmospheric grid resolution.



### 3.7 Positioning of gauging stations

Stream gauging stations are a precious tools to validate the simulated water cycle of land system models at catchment scale. For this reason, it is important to be able to localize these stations in the HTU space. Obviously, depending on the fidelity of the river graph in the HTU space more or less stations can be reliably placed. The position of the stations will be made according to their geographic position and the error in the upstream area within the model. The user can choose the maximum error in distance and fraction of upstream area which can be tolerated.

After the construction of the HTU graph, the pre-processor will attempt to place each stations within the tolerance selected by the user. The errors will then be minimized to select the HTU which will be considered representative for the given station. This information will then be archived with the HTU network so that the land surface model can monitor the flow out of the HTU corresponding to the station during the simulation. As the stations are placed in the HTU space, when the characteristics of the graph change more or less stations can be positioned within the allowed errors.

## 4 Validation of HTU graph

A method is needed in order to statistically validate the quality of the HTU graph and identify the deviation from the original HDEM induced by the reduction of effective resolution operated by the algorithm described above. As HTU graph construction is designed to work without human supervision and at global scale, a visual inspection is not sufficient.

The validation method samples randomly a large number of river segments which are representative of the network. The
length and elevation changes for each of these segments are computed on the HDEM and on the HTU graph to evaluate the errors. The red line in Figures 2c) and d) represent one such segment. Its length and elevation change computed on the HDEM by summing over all pixels it traverses is our reference. In HTU space, the river segment is represented by the blue lines and illustrates the large differences with the select truncation ($nbmax = 18$ and 10 in Figures 2). The properties of the stream within the HTU is given by the white arrows within the area to which the outflow points belongs (represented by a grey
circle). The properties of the segment in HTU space can thus be computed by summing the HTU's stream length and elevation change along the blue lines. By comparing it to the reference the degradation of the stream properties in the HTU space can be estimated.

The errors of segment properties can be decomposed into a cell and a topological error. Within each HTU we can compare the sub-segment's properties to the one of the HTU. This will be called the cellular error. In Figure 2c) the properties of HTU
8 are given by the white arrows and they are different from those of the red line. We will thus have a small cellular error here. In the downstream HTU (number 4) on the other hand, the cellular error is zero as the segment overlaps with the main river (white arrows). The topological error describes differences in the path followed by the rivers. In Figure 2c) at $nbmax = 18$ the river La Barberolle passes through Valence on its way into the Rhone. For $nbmax = 10$ (Figure 2d)) this river flows into another one south of the city before joining the Rhone. This is a topological error which is independent of the properties of
each of the HTUs. As this error is difficult to quantify, it will be estimated by subtracting the cellular error from the total error.



This methodology is applied over the largest European rivers (Danube, Rhone, Rhine, Loire and Elbe) for the four atmospheric grids presented above. The relative errors and their statistics are computed over a sample of 8000 segments. To ensure a good representativeness the length of the segments is chosen between $100$ and $1000\,km$. To avoid unrepresentative starting points, their upstream area should be at least of $10\,km^2$. Other values for these parameters of the sampling method were tested but it did not change the results.

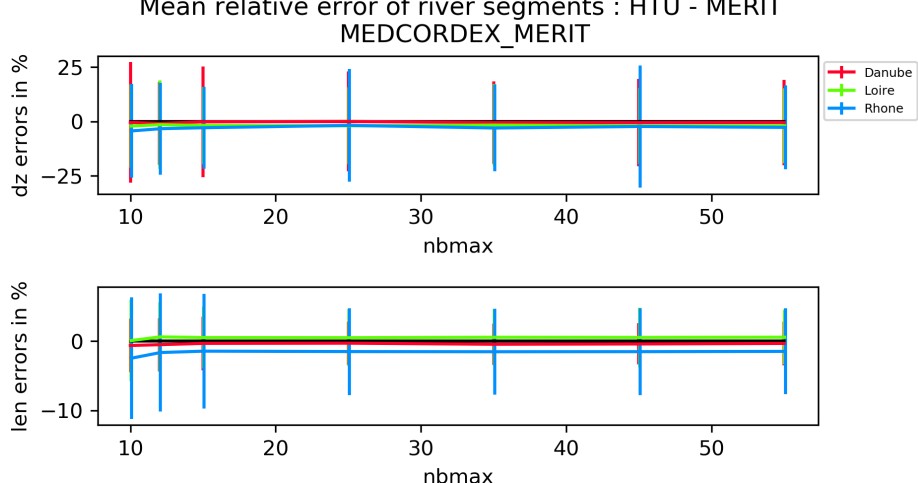

**Figure 3.** Figure provides the distribution of errors for the elevation change (dz) and length (len) for the 8000 samples over 7 different truncations. The error bars represent the standard deviation of the error sample.

Figure 3 shows the dependence of the average relative errors with the chosen truncation ($nbmax$). The error bars in the figure provide the variance of relative errors within the sample of river segments. The variance is driven by the diversity of segments drawn in terms of length (affecting the relative errors) and the complexity of the hydrological structure. This analysis is only displayed over 3 basins not to clutter the graphics too much. For the length of the segments the errors are within a range $\pm 10\%$ with the largest standard deviation found for the smallest truncations. For elevation change most relative errors are smaller than $25\%$ with again the largest spread obtained for $nbmax = 10$. The large variance of errors around the mean imposes that variations of the mean relative errors need to be tested statistically. To this end the mean errors are compared to the next higher truncation using a simple t-test for two independent samples. This will allow to show if increasing the number of HTUs used to represent the rivers significantly improves the quality of the graph or not.

Let us now analyse how the quality of the HTU graph depends on the resolution of the atmospheric grid and the maximum number of HTUs allowed per grid ($nbmax$). First we examine in figure 4a) the evolution of the mean relative error with increasing $nbmax$ on the coarsest grid (WFDEI). For the lowest truncation ($nbmax = 10$) on the Rhine the mean segment elevations changes ($dz$) are lower than those in the HDEM by about $10\%$. This error is larger at the cell level (dashed lines), meaning that on average the $dz$ of the HTUs are smaller than the corresponding sub-segment by over $15\%$. This difference of







**Figure 4.** Error decomposition for the segment's elevation changes and length for 5 rivers at 3 different atmospheric resolutions and two different HDEMs





errors is explained by the fact that on average the connections between HTUs are further downstream than on the HDEM, thus
compensating partly the cellular error. This large difference between the total and cell mean errors is not an ideal solution as it
means that the actual course of the river is not well respected by the cascade of HTUs. The same behaviour is also found for
the length of the river segments with an even larger compensation by the topology of the river. Figure 4a) also demonstrates
that as $nbmax$ increases the total mean error reduces and is overwhelmingly explained by the cell error. For the WFDEI grid

the optimum is obtained for $nbmax = 35$. Above this value increasing truncation does not significantly change the mean error
any more according to the t-test. At this point we have enough HTUs per $0.5°$ grid box that the only error in the selected river
segments are those linked to the fact that the HTU's properties are slightly different from the elements of the segments crossing
them. It can also be noted that from this truncation onward the mean relative errors in elevation changes and segment length
are stable and small.

Figure 4c) shows the same results for the highest resolution considered here ($11\,km$). The mean relative errors in $dz$ and $len$
are smaller than the coarser grid considered above, also for high $nbmax$ values. It also demonstrates that the convergence of
the cell and total relative errors occurs already at $nbmax = 15$. The higher atmospheric resolution facilitates the aggregation
of the hydrological information of the MERIT HDEM which is at $60\,arcsec$ resolution. There are about 8 grid cells of the
EuroMED grid in each WFDI grid. But we can obtain similar qualities for the HTU graph in WFDEI with only a little more

than twice the number of HTUs. This shows that the algorithm to aggregate the hydrological information at the atmospheric
grid captures well the main features which determine the flow of water along the network. Obviously, would the HDEM be of
higher resolution (MERIT is available at 90m) then also for the EuroMED grid more HTUs would be needed to describe the
network.

    For the MEDCORDEX grid at $20\,km$ we have build the HTU network with two different HDEMs. For the MERIT HDEM

($60\,arcsec$ resolution) the convergence of cell and total relative errors occurs at $nbmax = 25$ and are better than $5\%$ for both
the elevation change and length of samples (Figure 4b). When HydroSHEDS ($30\,arcsec$ resolution) is used the behaviour
changes slightly. Let us first examine the errors for the length. The convergence of errors occurs at $nbmax = 35$. Because of
the higher resolved hydrological information more HTUs are required for its proper representation. For the segment length,
the mean errors are similar than for the MERIT HDEM. The situation is more complex for the elevation changes of the

segments. Because HydroSHEDS does not provide a hydrologically corrected topography, the errors are much larger than
for MERIT (Yamazaki et al., 2012). When following downstream the segments in HydroSHEDS a number of situations are
encountered where the elevation change is negative, i.e. water flow into a pixel with a higher elevation. For shorter segments
this occurs for about 20% of the pixels while for longer segments, which are more likely to traverse flat areas, it can reach
30%. On the HDEM, for the full segment, these errors compensate and the total elevation change can be assumed to be correct.

But when decomposing it into pieces the compensations can be interrupted. Furthermore when constructing the HTUs we have
imposed that $dz$ cannot be smaller than $0.1\,m$ thus creating a large discrepancy with the segment as defined in the HDEM.
Figure 4d) shows that for rivers with large flat areas such as the Danube the error is largest while it is relatively small for
catchments dominated by mountainous areas such as the Rhone.




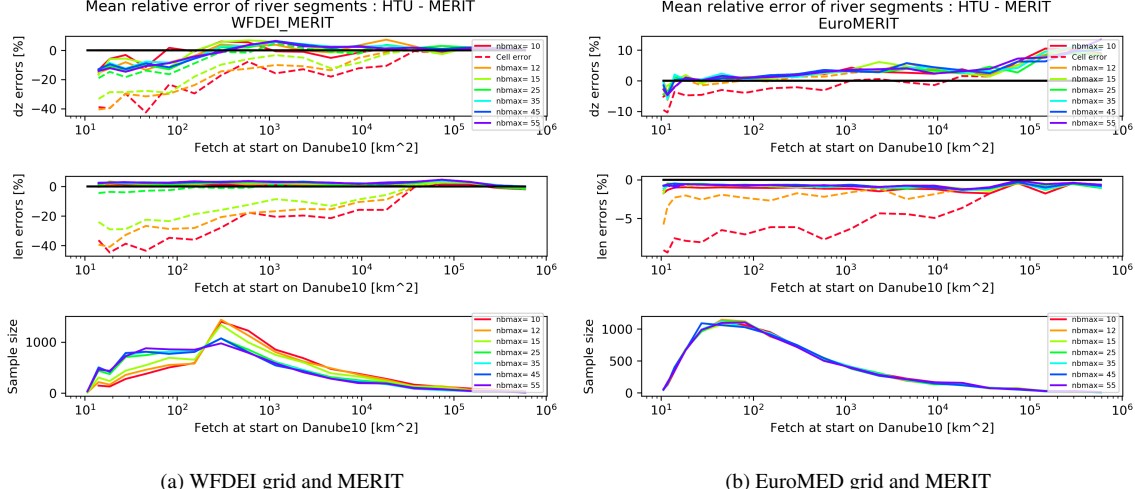

(a) WFDEI grid and MERIT                         (b) EuroMED grid and MERIT

**Figure 5.** Distribution of errors along the upstream area (or fetch) of the starting point of the segments for the Danube.

To determine the main sources of errors in our samples, their distributions relative to the upstream area of the initial point
are analysed for the Danube. Figure 5 displays these distributions for the coarsest and finest grids analysed here. First it has to
be observed that because of the dominance of small sub-catchments in any river basin, the sample of segments favours starting
points with about $100\,km^2$ of fetch. Again we can see for which $nbmax$ the cell and total errors converge but also that the
higher truncations are less needed for larger basins as the convergence there is faster. The mean error in segment length is stable
with changing upstream area. On the coarse grid we can note that for small rivers (fetch $< 300m^2$) the $dz$ error is relatively
large. This error is amplified by sampling bias and thus has a major contribution to the mean error. This error in $dz$ for small
catchments is consistent with the hypothesis of the decomposition algorithm which favours the properties of the largest river
in a grid box to guide the graph construction. This leads to small catchments having a higher probability to display segment
properties that do not match those of the HTU.

It must be noted that for the EuroMED grid the error in $dz$ increases with the size of the upstream area. This is simply
because segments with large upstream areas are more likely to cross flat areas of the Danube basin. The distribution of segment
errors was also computed against the length of the sample (not shown). For both the length and elevation change errors it is
found that they are quite constant except for short segments. As they are likely to have also small upstream areas they display
a relatively large cell error.

In conclusion of this section we provide the optimal truncation for each grid used here (Table 3). The values for $nbmax$ are
derived from Figure 4 as the point where reductions in errors in the graph are not any more justified by the extra computational
cost.



**Table 3.** The atmospheric grids and land/sea masks for which HTU graphs were built to test the methodology. If not specified otherwise the MERIT HDEM is used.

| grid name | Resolution | Optimal truncation for MERIT ($nbmax$) | Optimal time step [$s$] |
|-----------|------------|----------------------------------------|--------------------------|
| WFDEI | $0.5°$ | 35 | 1800 |
| E2OFD | $0.25°$ | 15 | 900 |
| MEDCORDEX | $20\,km$ | 12 | 900 |
| | | HydroSHEDS = 25 | 450 |
| EuroCORDEX | $11\,km$ | 10 | 450 |

## 5   Numerical implementation

For the constructed HTU graphs the constants $g_X$, which are the inverse of a velocity, need to be estimated. Tests with OR-CHIDEE have shown that for the two high resolution HDEMs the same values can be used and for the coarser one the constant
for the stream needs to be changed slightly (see Table 2)(Schrapffer, 2022). It is quite likely that these parameters are more dependent on the land surface scheme used here than the HDEM. The repartition of water fluxes between surface runoff ($R_{\hat{i}}$) and drainage ($D_{\hat{i}}$) as well as the inertia of the soil moisture model will play a key role for these constants. Thus the values given in Table 2 are probably only valid for ORCHIDEE's current soil moisture model (de Rosnay et al., 2002; d'Orgeval et al., 2008) in its $2m$ depth configuration. Using the HTU graphs produced here in any other land surface model will probably
require to readjust these parameters.

The water continuity (Eq 5) is discretized in time using an explicit numerical scheme. This means that the fluxes $Q_{j,X}$ are evaluated before the reservoir content ($W_{i,X}$) are updated. This choice simplifies the implementation and the interaction with the parallelisation of the ORCHIDEE code. The heat transport equation 9 is also implemented using an explicit method. In order for the relaxation term to be able to efficiently avoid the singularity of empty reservoirs it is evaluated implicitly. This
flux will thus depend on the current upper soil temperature and the stream temperature at the next time step.

The Courant–Friedrichs–Lewy (CFL) condition mandates that for a convergence of the numerical solution the time step needs to be smaller or equal to the residence time ($\lambda_{i,X} g_{i,X}$) of the water in the fastest reservoir of the HTU : the stream. Given the decomposition of the atmospheric grid into HTUs for a finer representation of the hydrological connectivity, a wide distribution of residence times will be obtained with some values being very short. It is thus not practical to select a time step
for the routing scheme based on the smallest residence time on the domain. The scheme thus needs to be able to cope with unstable flux calculations. To this end the numerical solution for the continuity equation (Eq 5) also includes a flux limiter given by : $Q_{j,X} \leq W_{i,X} \Delta t$. This condition should only activate for reservoirs with low water contents and short residence times. As this flux limiter also determines the numerical quality of the simulated discharge, the model monitors how often this condition has to be imposed.

A practical solution for choosing the time step of the routing scheme is to select a position within the area weighted distribution of residence times of all HTUs within the computational domain. It is proposed to select the time step corresponding





to the 25% quantile of this distribution. This means that 75% of all HTU by area will not violate the CFL criteria while for the others there is a risk that the solution will not be correct. As a consequence it needs to be verified that this compromise on the quality of the numerical solution on some HTUs will not affect the simulated discharge at the spatial scales of interest. We

found that the time steps determined with this method are more dependent on the atmospheric resolution than the hydrological truncation. This is quite convenient as with a refining of the atmospheric grid the time step of the processes which have the land surface scheme as a lower boundary will also decrease. Thus increasing the frequency at which the routing scheme will need to be called is not a strong constraint.

This section explores the convergence of the simulated discharge with the selected time step ($\Delta t$), the number of HTUs

($nbmax$) and the resolution of the atmospheric grid. The analysis is performed on a number of simulations over the Euro-Mediterranean domain discretized with the atmospheric grids presented in Table 1. To focus on the numerical properties of the routing scheme, a version decoupled from ORCHIDEE is used which is directly forced with daily mean surface runoff, drainage and temperature profiles extracted from a previous WFDEI-GPCC (Weedon et al., 2014) forced simulation of ORCHIDEE. These fields have been interpolated to the selected time-step for the routing and to the spatial resolution of the atmospheric grid

on which the routing scheme is to be evaluated.

These simulations are evaluated at a limited number of gauging stations which cover a range of upstream areas and climates. From the over 3800 stations placed on the river graphs of the Euro-Mediterranean region only a few were selected with upstream areas ranging from 2500 to $8\,10^5\,km^2$. The lower limit is given by the spatial resolution of the WFDEI forcing and ensures that the smallest catchments contain at least one atmospheric grid box. The largest catchment is the one of the Danube.

The list of these stations is provided in Appendix A. At each of these points in the graph, the simulated river discharge and temperature are evaluated against a reference configuration. The metrics used to quantify the change of behaviour of the model are the Nash–Sutcliffe model efficiency (NSE), the correlation and ratio of standard deviation. These metrics are computed with daily values over an 11 year period going from 1983 to 1993.

### 5.1 Role of the time step

The length of the time step is first tested on the graph produced with MERIT for the WFDEI grid using $nbmax = 35$ based on the evaluation in section 4. The time step is progressively increased from $225\,s$ to $5\,h$ and the smallest value is considered to be our reference.

As a first step, we examine the fraction of HTUs where the flux limiter has to be applied to stream flow. As this variable is quite constant throughout the period analysed, only the annual mean is shown in figure 6 with the selected stations ordered by

up-stream area along the x-axis. In the y direction the tested time steps are plotted and each dot measures the quality of one simulation compared to the reference. This graphic demonstrates that the flux limiter applies to all catchments independently of their size but the frequency of its activation is a strong function of the time step used. The horizontal line shows the time step at which 75% of HTU satisfy the CFL criteria. At this level the flux limiter has to be applied to less than 20% of the HTUs. Thus, when looking at the convergence of the solution with the time step we have to consider that not only the CFL criteria

deteriorates the solution but also the flux limiter.





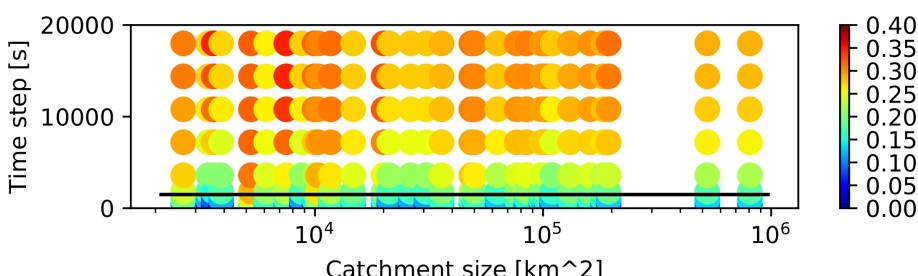

**Figure 6.** At each station and for the range of time steps tested, the fraction of HTUs for which the flux limiter has been activated in the upstream catchment.

Figure 7 compares for the same set of experiments but now examining the degradation of discharge and stream temperature with time step when compared to the shortest value ($\Delta t = 225\,s$) using the three metrics discussed above. The Nash–Sutcliffe model efficiency is close to one for all simulations performed with time steps close to or lower to the recommended value (horizontal line). The correlation between the reference and the test simulation remains close to one as well as the ratio

of variance. In this range of time steps we can consider that the numerical solution has converged for catchments larger than $2500\,km^2$. Once the 25% quantile of residence times is passed some stations show quick degradation in the simulated discharge and we can note that generally small catchments are more sensitive than larger ones. On the most downstream stations of the Danube (to the right of the x-axis) the degradation only becomes apparent for time steps of a few hours. For smaller catchments this can occur much earlier. This degradation is not a linear function of up-stream area as it also depends on the complexity

of the river graph and the topography. The degradation of the river discharge is manifested by a reduction of the correlation of daily discharge values with respect to the reference and a decrease of variance. The peak discharges are more strongly affected by increasing time steps than the base flows.

The right column in Figure 7 shows that stream temperature is much less sensitive than discharge to increasing time steps. This is not surprising given that the rivers have relatively slow fluctuating temperatures. The mass flux is thus the driving

constraint for the temporal discretization of the routing scheme.

Figure 8 shows only the NSE metric but now for three atmospheric grids and in one case for HTU graphs built with a different HDEM (HydroSHEDS). The results are very similar to the ones presented above. We can conclude that the 25% quantile of the residence time is a quite robust method to determine the time step which satisfies the CFL criteria and keeps the flux limiter to an acceptable level. Comparing the figures in the upper row shows that the criteria recommends a shorter time step

for the HydroSHEDS HDEM (See table 3) as it provides hydrological information at $30\,arcsec$ resolution. The degradation of the comparison to the reference occurs for longer time steps than for MERIT although we are on the same atmospheric grid (MEDCORDEX). This can probably be traced back to the quality of the hydrological information provided by both HDEMs. In both cases the methodology for selecting the time step seems to be too conservative on this $20\,km$ grid. This results is



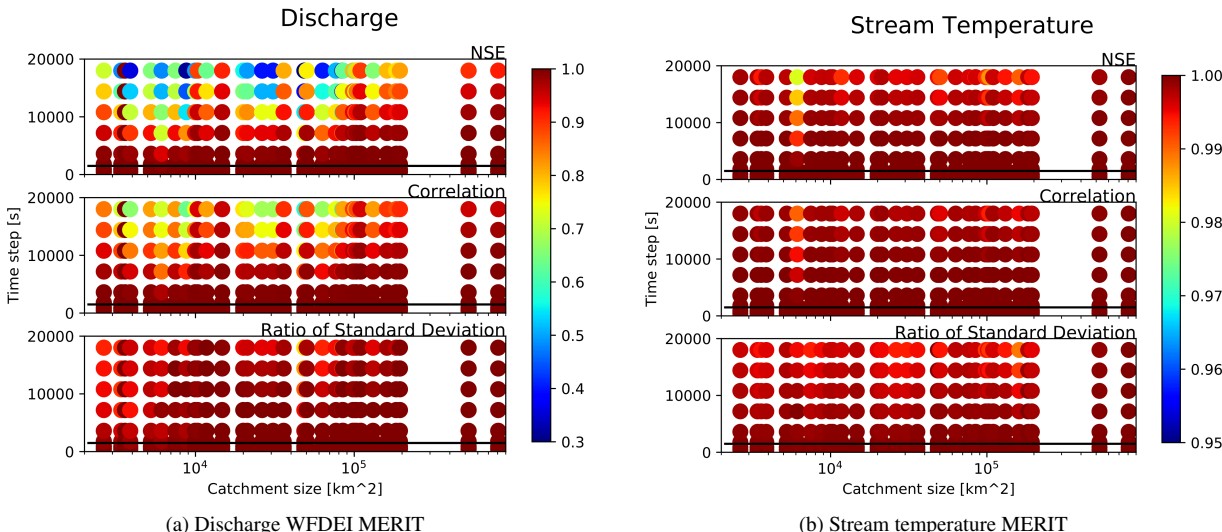

Figure 7. Comparing simulations on the WFDEI grid (with the MERIT HDEM and $nbmax$=35) with increasing time step to the reference at $\Delta t = 225\,s$.

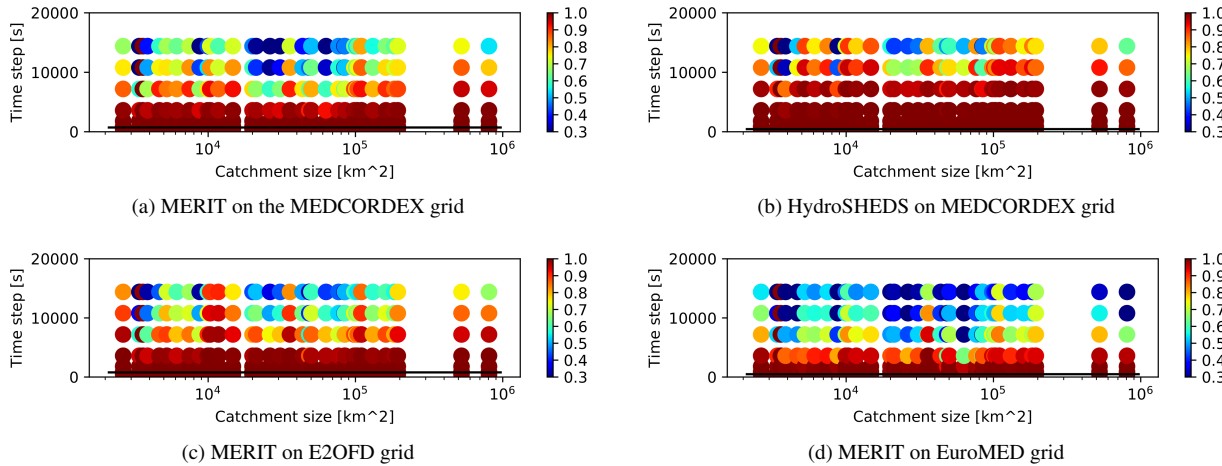

Figure 8. Convergence of simulated discharge with increasing time steps on the four different grids presented in Table 1 and the two HDEM for the MEDCORDEX grid. Only the Nash-Sutcliffe efficiency is displayed.

confirmed on the E2OFD and EuroMED grids as the degradation of the numerical solution occurs for time steps larger than the recommended value. But it has to kept in mind that the runoff forcing used here is an extrapolation of data from a $0.5°$ grid and thus it is of lower temporal and spatial variability than one would expect at the higher resolutions presented here. Furthermore on the higher resolution grids, given the appropriate atmospheric forcing, the user would want to analyse smaller catchments and more extreme hydrological events.





## 5.2 Role of truncation in HTU space

The same type of analysis can be performed to explore the impact of the number of HTUs used per grid box on the time step. For each case we use the time step proposed by the 25% quantile of the residence time distribution. As the discharge is evaluated at gauging stations, when $nbmax$ is modified the position of the station within the graph can also be affected. Thus the results presented here not only evaluate the impact of the truncation on the simulated discharge but also the positioning of the sampled point. These two factors are not independent and both affect our ability to reproduce observed river flows.

The y-axis of Figure 9 now shows the maximum number of HTUs per atmospheric grid ranging from 10 to 45. The configuration $nbmax = 55$ is considered as our reference here. The graphics show that the truncation is of less consequence to the quality of stream levels and temperature, within the range considered here, than the time step. For the smallest value of $nbmax$ the NSE only falls to 0.75. In none of the three metrics considered here there is an indication that the result depends on catchment size. If one were to consider maximum number of HTUs lower than 8, then not all outflow directions of the rectangular atmospheric grids can be represented and the impact on the simulation would probably be more severe. These extremely low values of $nbmax$ were not considered as they defeat the point of a sub-grid approach to routing.

All stations where a significant degradation of the simulation occur, it takes place below $nbmax = 35$. This result is consistent with the statistical analysis of the river graphs presented above (Section 4). For MERIT on the WFDEI grid, the quality of the segments starts to decrease for $nbmax$ below 30. This point is not as clearly visible here as the analysed stations all have a relatively large up-stream area. As demonstrated with Figure 5 the errors in the river segments are more pronounced for catchments smaller than $10^4\,km^2$. To demonstrate the loss of numerical convergence for discharge caused by a coarse HTU graph stations in small catchments would need to be sampled. But this would not be consistent with a forcing at $0.5°$ resolution.

## 5.3 Role of the resolution of the atmospheric grid

Based on the analysis of the graphs presented in Section 4 an optimal value for $nbmax$ and time step has been selected for each combination of atmospheric grid and HDEM (See Table 3). The objective here is to determine how sensitive the simulated discharge is to the atmospheric grid given the same runoff and discharge fields. As they originate from a WFDEI simulation on the $0.5°$ grid this will be used as our reference here.

Figure 10 shows for the selected stations the deviation of the three other configurations from the WFDEI-MERIT case. One can note that the changes in the three metrics used are generally small. There is no systematic dependence of the numerical quality with the grid or catchment size even though the resolutions and projections are quite different. One may note that there is systematic degradation of the NSE efficiency, correlation and ratio of standard deviation when changing the HDME on the MEDCORDEX grid. The MEDCORDEXHS simulation differ particularly for stations with a catchment larger than $7\,10^4\,km^2$. This can be expected as it is based on another description of the topography and river flow directions than the reference. Difference accumulate with the size of the basin. When only the resolution changes (E2OFD, MEDCORDEX, EuroMED), the impact on the simulated variables is low except for a few stations. For the E2OFD configuration the station Rekingen on the



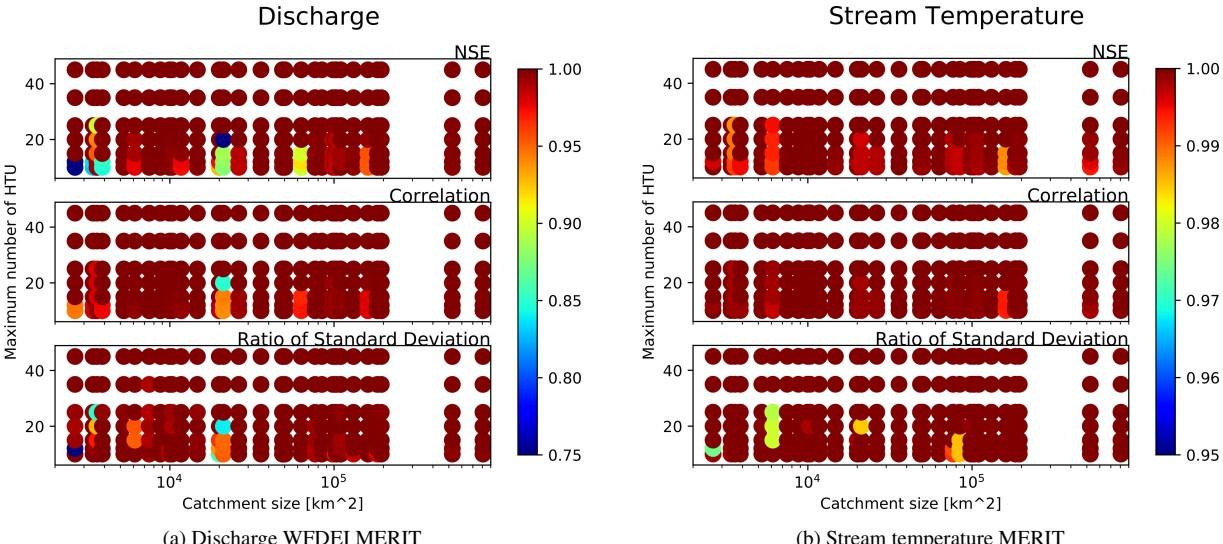

(a) Discharge WFDEI MERIT                              (b) Stream temperature MERIT

**Figure 9.** Simulations on the WFDEI grid (with the MERIT HDEM and $\Delta t = 900\,s$) with increasing maximum number of HTU are compared to the reference which is chosen to have $nbmax = 55$.

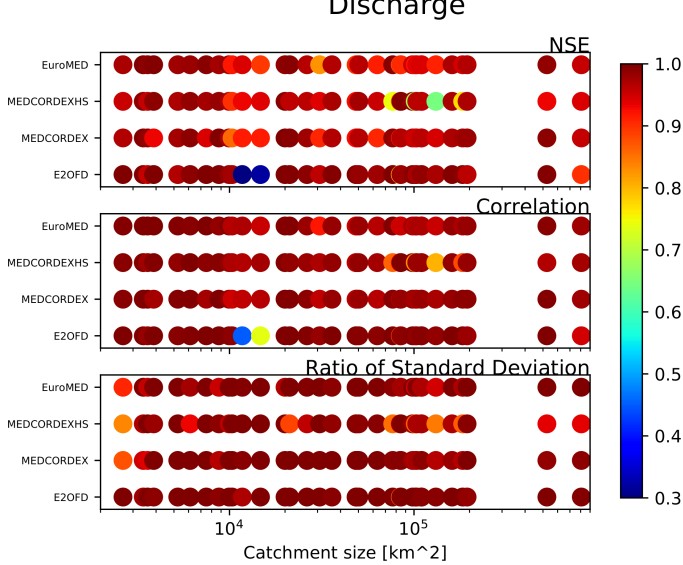

**Figure 10.** The impact of the atmospheric grid is evaluated by comparing the WFDEI simulation to the three other grids (E2OFD, EuroMED and MEDCORDEX) and for one grid also another HDEM : MEDCORDEXHS (See Table 3).

Rhine (catchment area of $1.4\,10^4\,km^2$) stands out. As it is mostly the correlation and the NSE which are affected it has to be assumed that it is positioned differently in the river graph.





### 5.4 Convergence of the numerical solutions

It is an important result that for an optimal truncation and time step selected according to the criteria defined above, the
simulated stream flows and temperatures are relatively insensitive to the atmospheric grid on which the climatic forcing is
provided. This removes the necessity to adjust the parameters $g_X$ of the routing scheme when running at different resolutions.

Nevertheless, some caveats have to be kept in mind. The numerical verifications presented above were performed using
output from an ORCHIDEE simulation forced by the WFDEI-GPCC forcing at $0.5°$ resolution. Because of its 3 hourly forcing
there is little diurnal variability in the surface runoff and generally the spatial variance is limited by the coarse atmospheric
information. The higher resolutions evaluated here were run with variables which do not include all the spatial and temporal
variability which would be expected. Thus, when using the routing at high resolution within regional Earth system models the
criteria for selecting the time step might need to be reduced. In particular when moving to convection permitting resolutions
when the intensity of simulated rainfall will modify the behaviour of the land surface model and thus the dynamics of the fluxes
which feed river flows. It is nevertheless reassuring to see in the analysis presented above that for higher resolutions a margin
exists for the time step.

Once reliable km-scale atmospheric forcing are available the numerical analysis presented here should be revisited with
a particular attention to flood events in small catchments. It will allow to evaluate at which stage a smaller quantile in the
residence time should be selected and how important this choice is for representing the extreme hydrological events we are
interested in.

### 485  6   Validation of the routing scheme in ORCHIDEE

In order to appreciate the importance of the numerical choices presented above, the routing scheme is evaluated jointly with
ORCHIDEE at two different resolutions but only using the MERIT HDEM. WFDEI and E2OFD (Marthews et al., 2020) are
two state of the art atmospheric forcing for land surface models both based on the ERA-I re-analysis (Weedon et al., 2014).
With the advent of higher resolution precipitation products it was decided to interpolate WFDEI to the MSWEP (Beck et al.,
2017) grid ($0.25°$) and use it to bias correct rainfall from the re-analysis. Thus the two driving data are not only available on
different atmospheric grids but also differ in the precipitation intensity and spatial distribution. The WFDEI version which was
bias corrected with GPCC (Becker et al., 2013; Schneider et al., 2014) is used here.

Figure 11 shows on a Taylor diagram the quality of the simulated discharge when compared to observations at the 35
stations presented in appendix A. For discharge, the database of GRDC (Fekete et al., 2000) (http://grdc.bafg.de) is used.
It has been enhanced with national sources of information for a number of European countries. The stream temperatures
were obtained through the GESM database (Programme, 2017) (https://gemstat.org/). The diagnostics are computed with the
available observations over the period 1983-2002.

For discharge these results are well in-line with previous validations of ORCHIDEE (Ngo-Duc et al., 2005) and the experi-
ence gained during the data assimilation experiments performed (Wang et al., 2018). The discrepancies between the models
in the Euro-Mediterranean region can be attributed to missing processes in the land surface model. In this region in particular,





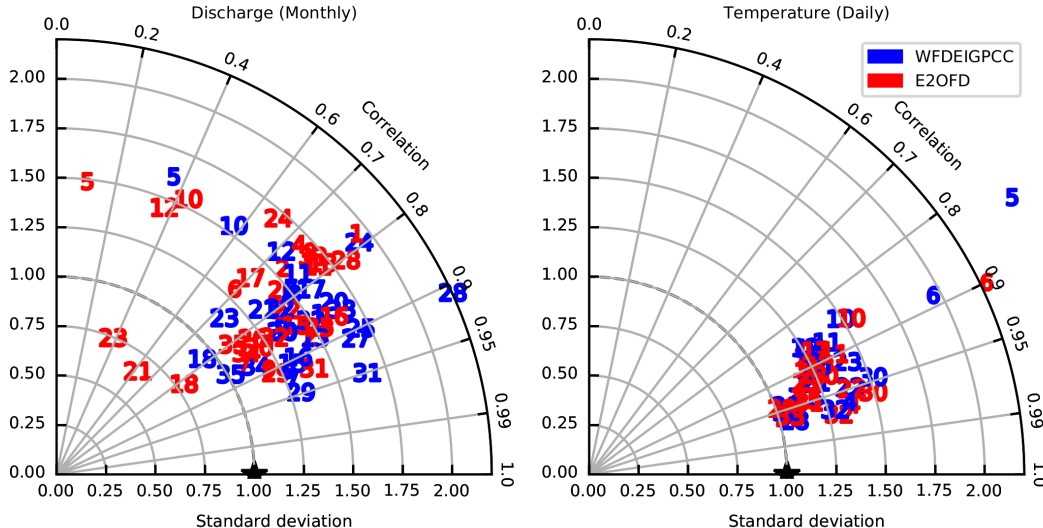

**Figure 11.** Taylor diagrams comparing the observed monthly discharge and daily stream temperature to observations available at the 35 stations listed in Table A1. The numbers indicated in the figures correspond to those in the table.

human management of surface water is missing as it has not yet been included in the version of the model used here (Zhou et al., 2021). The differences between both simulations are of more interest here. Based on the analysis above, we know that if the forcing are the same, the correlation of discharge is better than 0.9 (except for stations 10 and 11) and the ratio of standard deviation is close to 1. These metrics can be transferred to the Taylor diagram to see that the difference in disagreement of

both simulations with observations cannot be explained by the numerical scheme. This confirms that the difference of imposed atmospheric conditions is the dominant cause of differences in behaviour of the model (Marthews et al., 2020).

More original for ORCHIDEE is to verify the quality of the simulated stream temperature. The Taylor diagram might give the reassuring impression that the relatively simplistic model used here for temperature is satisfactory. First and foremost we have to remember that stream temperature is less affected by water management than water levels. The temperature in streams

is driven by the seasonal cycle of its values over land surfaces and thus the temporal correlations can be expected to be high. On the ratio of standard deviation it can already be noted that the amplitude of the stream temperature is overestimated with the simple model proposed here. The stations of Porte du Scex and Diepoldsau (Stations 5 and 6 respectively) are outliers and will be considered in more detail in the next section.

To better understand the qualities and limitations of the simple temperature scheme proposed here let us examine two stations

with a large upstream area and an excellent observational record : Lobith on the Rhine and Nagymaros on the Danube (Stations 31 and 32 respectively). These stations where also used as validation points by models which attempt to incorporate a much wider set of processes governing stream temperature (van Vliet et al., 2012; Tokuda et al., 2019; Liu et al., 2020). One notes that on the Rhine the model has a systematic cold bias (Figure 12). Because of the strongly underestimated winter temperatures





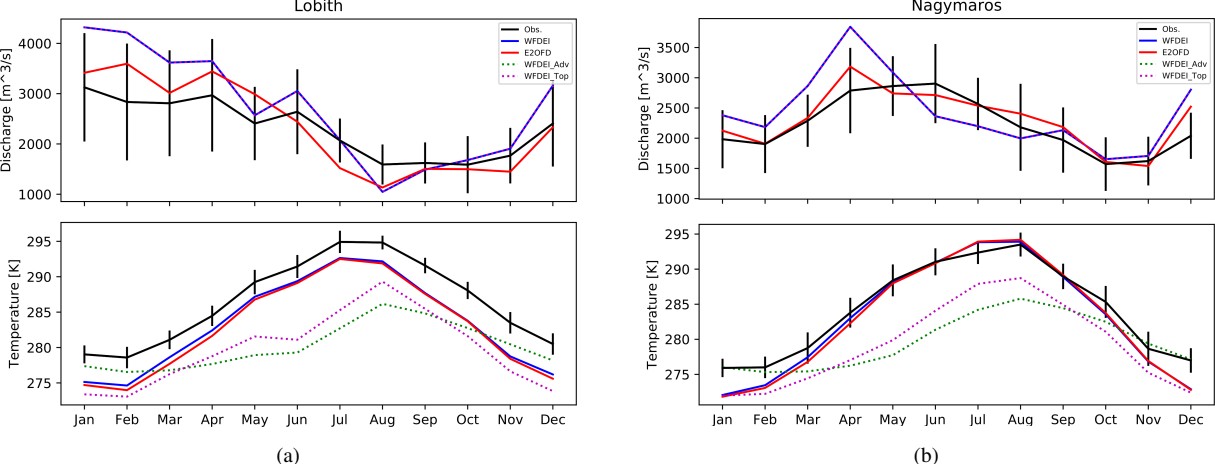

**Figure 12.** Mean annual cycle of river discharge and temperature at the Lobith station on the Rhine and Nagymaros on the Danube. For the observations the inter-annual variability of the monthly values is displayed as error bars. The two reference configurations (WFDEI and E2OFD) are drawn with plain colors while the two sensitivity experiments carried out with WFDEI are given as dotted lines.

($-5\,K$), the amplitude of the annual cycle is too large. On the Danube the annual cycle of temperature is closer to observations.
But again, the strong cold bias in winter exaggerates the amplitude of the annual cycle. The underestimated winter temperatures is something which was also found in the HEAT-LINK model on the Danube (Tokuda et al., 2019) while the VIC model (van Vliet et al., 2012) had stream temperatures for the Rhine which were overestimated in winter. Another striking feature is that all three models predict a too rapid cooling of the rivers in autumn.

Figure 13 confirms that this diagnostic is valid over the 25 stations with a temperature record and is general over the region.
The two reference simulations show a systematic underestimation of stream temperature in winter and relatively reliable summer values. These biases do not depend on the size of the catchment and are larger than the differences between the forcing data. Thus there is room for improvement. It has to be remembered that the soil temperatures over the top $0.3\,m$ are essentially driven by the land surface temperature (LST) simulated by ORCHIDEE. This variable is well known to be affected by modelling assumptions and errors in the driving data (Huntingford et al., 2000). Previous studies have shown that
reproducing remote sensed LST in land surface models is challenging (Barella-Ortiz et al., 2017).

Given that our approach is relatively simple, it is feasible to better understand the origin of these biases in the simulated annual cycle of stream temperature.

## 6.1 Sensitivity of the stream temperature to model assumptions

Stream temperature is determined by two fundamental drivers :





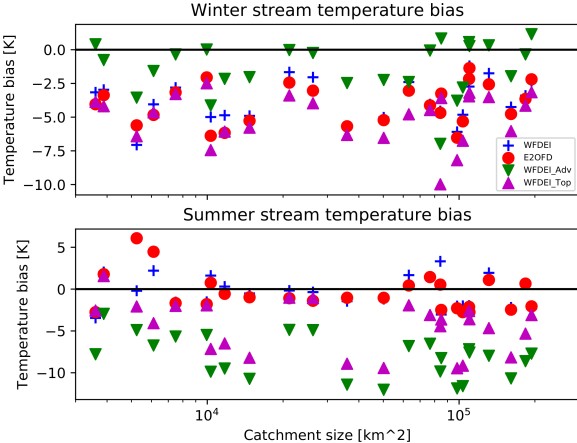

**Figure 13.** Seasonal biases at 25 stations where stream temperature are available for the two reference simulations (blue + symbols DEI and red dots for E2OFD) and for both sensitivity experiments (green downward triangles for the pure advection of both temperatures and purple upward triangles for top temperature advection).

– the temperature at which the water leaves the ground and often referred to as headwater temperature. For instance, van Vliet et al. (2012) compared an empirical relation for the headwater temperature with the case when daily soil temperature is chosen, without specifying at which depth. Tokuda et al. (2019) use the temperature of runoff water without specifying how it is estimated in MATSIRO (Takata et al., 2003). Here it is proposed to add to this boundary condition the distinction between temperature of surface runoff and deep drainage temperatures (Section 2.2).

– The energy gained by the stream through interaction with the atmosphere and the landscape as it flows to the ocean. These processes are represented with great detail in some models like HEAT-LINK and VIC. Here they are simplified to a very basic relaxation towards top ground temperature and thus a variable closely related to LST.

Separating these two factors in the deficiencies of the models is key to future developments of the scheme so that errors in the assumptions for the boundary conditions are not compensated by biases in the interactions of the river with the landscape and
atmosphere.

     To determine the largest source of error in the simulated stream temperature two sensitivity experiments are carried out with the WFDEI configuration in which the interaction with the atmosphere and landscape are suppressed by setting $a = 10^5$. This means that the relaxation term only serves its numerical purpose as it is different from zero only when the HTU water levels tend to zero. The two experiments proposed here are :

**WFDEI_Adv** Runoff and drainage have the temperatures as defined in Section 2.2.

**WFDEI_Top** Both water fluxes leaving the soil moisture scheme have the temperature of the top soil layers $(0 - 0.3\,m)$.



The fact that the interaction with the atmosphere and landscape is suppressed in both these experiments leads to a strong underestimation of the summer stream temperature (Figure 13). The two largest stations on the Danube and Rhine (Figure 12) show that without the additional energy brought to the stream by the environment the amplitude of their annual cycle is much lower. When the near surface temperature is used for both runoff and drainage, the amplitude increases slightly but it is still not at the correct level. Figure 13 shows that this result is valid for all catchment sizes. The differences between WFDEI and WFDEI_Adv do not systematically increase as the stations have a larger catchment. A conclusion which can be drawn at this stage is that if rivers only obtain their temperature from the headwaters they do not warm-up enough in summer. This demonstrates the importance of the exchanges with the atmosphere and landscape during this season. The relaxation to the top soil temperature used here is just a simple transfer of the energy balance performed within ORCHIDEE to the streams. In view of the importance of this process a more complete representation of these exchanges should be aimed for. This includes the simulation of lake thermodynamics (Bernus and Ottlé, 2022). It also explains why simple empirical relations relating near surface air temperature and stream temperature work so well (Ducharne, 2008).

More interesting for the model development is the dependence of the winter temperature bias on the assumption made for the headwater values. Figure 13 shows that the case when drainage has the deep soil temperature and no atmospheric interaction is allowed (WFDEI_Adv) the results are more realistic. In this region, winter is the period of low flows which are dominated by the groundwater contribution to rivers. Attributing to drainage the warmer deep soil temperature in this season is a step in the right direction. This is corroborated by the fact that the two stations where the winter errors are the largest (Port du Scex on the Rhone and Diepolsau on the Rhine), and which are upstream of major lakes, are at the outflow of the Alps. In these topologically complex regions the contribution of groundwater to the low flows is particularly important (Floriancic et al., 2019). Rain and melt water from the previous season transits through deep aquifers before reaching the rivers at the bottom of the valley. It has also to be considered that the rivers flowing out of the Alps are affected by hydropower generation. This has been shown to significantly warm the streams in winter (Fette et al., 2007).

It is expected that in winter the interaction with the landscape and the atmosphere will cool streams as they are warmer than their environment. This is also what is found here when comparing WFDEI and WFDEI_Adv. Thus, in order to have simultaneously the correct interaction with the atmosphere and correct headwater temperature, the drainage temperature should be warmer than the $3 - 17m$ assumed here. The comparison with WFDEI_Top shows that when cooler near surface soil temperatures are used the streams are indeed colder, even without atmospheric interactions. This reveals a strong limitation of the routing scheme proposed here. As we do not represent well ground water, we do not know its depth or its residence time at different horizons. Thus its temperature when it emerges at the surface cannot be correctly established. As wintertime cold bias is also present in the mid-latitude basins reported by Takata et al. (2003) and they have assumed initial temperature to be that of runoff, the cause is probably the same as in our configuration. This highlights the importance of temperature to validate the groundwater component simulated in land system models.

In the Alpine region, winter stream temperatures have been shown to warm more slowly than in other seasons (Michel et al., 2020). As it is attributed to groundwater processes, land system models will need to better simulate this slower component



of the aquifer in order to be able to reproduce the observed trends in stream temperature and predict future evolution when coupled to ESM.

## 7    Discussion and conclusion

This paper proposes a methodology to decompose any atmospheric grid into a graph of hydrological Transfer Units (HTU)
based on a digital elevation model which includes flow directions (HDEM). This network allows to perform a hybrid routing which combined a vector-based with a grid-based approach (Yamazaki et al., 2013). It combines the water balance simulated on the atmospheric cells with the higher hydrological resolution needed to predict the horizontal transfers at the surface.

The algorithm introduces a truncation parameter which determines the maximum number of HTU as chosen by the user. This allows for applications which do not require all the hydrological details of river flows to reduce the memory consumption
of their land surface model by specifying a lower number of sub-grid elements. With a statistical sampling of random river segments the error introduced by the aggregation of the HDEM information at the HTU level is quantified. We recommend to use a truncation which reduces the total average error of the segments to the cell level error. This ensures that the graphs are correctly connected and minimizes the topological error. The optimal truncation will depend on the resolution of the atmospheric grid and the HDEM used.

The graphs of HTU would be useless if the water continuity equation cannot be solved with a reasonable time step. We propose to use the 25% quantile of the area weighted residence time as an optimal time step for the routing scheme. This ensures that 75% of the HTUs by area are numerically stable. It yields a time step which is compatible with the one of the land surface model. We find that the time step varies more with the spatial resolution of the atmospheric grid than the truncation of the graph. For the four resolutions tested it is equal or larger as the one typically used for ORCHIDEE when coupled to
the atmosphere. The time step can probably be increased if the continuity equation on the sub-grid part of the graph is solved implicitly. This would increase the computational efficiency of the routing scheme as it could be called less frequently by the land surface scheme.

Any routing scheme will have adjustable parameters in order to determine the residence time in the aquifers and surface reservoirs. ORCHIDEE has one parameter per reservoir represented in the transport scheme. We find that changing the reso-
lution of the atmospheric grid or the truncation does not require to readjust these parameters. The simulated discharge are not affected in any significant way by the resolution changes. Tests comparing two km-scale HDEMs also show that the parameters were largely independent of the hydrological information used to build the HTU graph. This is a very important result as it means that the routing scheme can be used over a large range of configurations of the ESM without having to be adjusted. Thus expertise can be transferred from one resolution to the other. Our hypothesis is that the repartition between surface runoff and
drainage of excess water in the soil moisture reservoir is the main driver for the optimal values of the parameters. Thus as the representation of the unsaturated soils evolves in ORCHIDEE the parameters of the routing scheme will have to be verified.

To demonstrate the value of the hybrid routing scheme on a graph of HTU a simple temperature transport scheme is implemented. It is remarkable that it produces results comparable to much more elaborate schemes. We attribute this to the fact



that in summer the energy balance over open water is relatively close to the one simulated by ORCHIDEE and thus a simple
relaxation is a good first approximation. More importantly, the simplicity of the scheme allows to reveal the reason for the poor
performance in winter which is also found in more complex models. During this season the bias is caused by the boundary con-
dition of the temperature scheme. The water appears not to stay long or deep enough in the ground in order to reach the streams
with a sufficiently high temperature. It points to a limitation of the groundwater representation in the current formulation of the
routing scheme rather than the temperature scheme.

This demonstration of the numerical robustness of the hybrid routing scheme of ORCHIDEE is an excellent starting point
for future developments. In a companion paper (Schrapffer et al. 2022) it is shown how the elevation information within the
HTU can be used to predict their flooding and overflow into neighbouring HTUs. This allows to represent the functioning of
fluvial floodplains and their contribution to evaporation and vegetation functioning within ORCHIDEE.

Lakes, reservoirs or dams can be placed onto the HDEM. During the construction phase of the HTUs this information can
then be transferred and aggregated to the level of the hybrid routing scheme. It can then be used to predict water volumes and
functioning of these hydrological elements. This can then be combined with the lake energy balance model (Bernus and Ottlé,
2022) in order to represent consistently the lateral transport of water volumes and thermal energy over continents within ESMs.

The HTU graph can be enhanced with adduction networks. They link demand points like irrigated areas, power plants (for
hydro power generation or cooling of other plants) or domestic needs to the most appropriate river. Should the demands not be
satisfied locally they can be propagated upstream on the adjoint HTU graph so that water is released by the appropriate dam
(Zhou et al., 2021). The proposed vision of lateral water transports within Earth system is an extremely power tool which will
enable us to integrate in land surface models the impact of human water management on the hydrological system and predict
its interaction with the climate.

The representation of deeper groundwater will need more attention in the future developments of ORCHIDEE as recom-
mended by the community (Gleeson et al., 2021). We have noted that probably only a more realistic representation of ground-
water transport will enable to correctly simulate temperatures and water quality parameters in areas with complex hydrogeo-
logical structures. It will have to be seen if a specific HTU graph taking into account the hydrogeology needs to be implemented
or if a full three dimensional representation will be required (Maxwell and Miller, 2005).

*Code and data availability.* The code and its evolution are freely available on the GitLab of the French Research organisations : https://gitlab.
in2p3.fr/ipsl/lmd/intro/routingpp. The hydrological digital elevation models used will be made available upon request.The code version and
data used for this study are available at https://doi.org/10.5281/zenodo.7058895

*Acknowledgements.* GRDC and WCRP are thanked for collecting and distributing the discharge and stream temperature data. We are greatful
to Dai Yamazaki for providing the MERIT hydrological digital elevation model. Without UKCEH's hospitality the lead author of this study
would not have found the time to perform the work. IPSL's MesoCentre is thanked for the computer time. ECOS-Sud has funded the



exchanges with the CIMA in Argentina. IPSL-Climate Graduate School-EUR (ANR-11-IDEX-0004-17-EURE-0006) is thanked for their support to the model development presented here.

Author contributions JP developped the code, designed and executed the numerical evaluations, AS contributed to the code development and evaluation, ED and OB contributed the temperature advection development, LR and JS evaluated the scheme during its development, XZ contirbuted to the code development and provided the MERIT HDEM, all co-authors discussed

the methodology and contributed to the manuscript.



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

**Appendix A: Stations used for analysis**



| Number | River | Station name | up-stream area $[km^2]$ | Available observations Discharge | Available observations Temperature |
|---|---|---|---|---|---|
| 1 | Danube River | Hundersingen | 2647.01 | 1983-2002 | - |
| 2 | La Seine | Troyes | 3410.00 | 1983-2002 | - |
| 3 | Vechte | Vechterweerd | 3600.00 | - | 1983-1995 |
| 4 | La Seine | Mery-sur-seine | 3880.00 | 1983-2002 | 1983-1996 |
| 5 | Rhone | Porte Du Scex | 5244.00 | 1983-2002 | 1983-2002 |
| 6 | Rhine River | Diepoldsau, Rietbruecke | 6119.000 | 1983-2002 | 1983-2002 |
| 7 | Trent, River | Colwick | 7486.00 | 1983-2002 | 1983-2002 |
| 8 | La Seine | Pont-sur-seine | 8760.00 | 1983-2002 | - |
| 9 | Thames, River | Kingston | 9948.00 | 1983-2002 | 1983-2002 |
| 10 | Rhone | Chancy, Aux Ripes | 10323.00 | 1983-2002 | 1986-2002 |
| 11 | Aare | Brugg | 11726.00 | 1983-2002 | 1983-2002 |
| 12 | Rhine River | Rekingen | 14718.00 | 1983-2002 | 1983-2002 |
| 13 | Danube River | Ingolstadt | 20001.00 | 1983-2002 | - |
| 14 | La Seine | Montereau-fault-yonne | 21178.00 | - | 1983-1996 |
| 15 | La Seine | Saint-fargeau-ponthierry | 26290.00 | 1999-2002 | 1983-1996 |
| 16 | La Seine | Alfortville | 30800.00 | 1983-2002 | - |
| 17 | Rhine River | Basel, Schifflaende | 35905.00 | 1983-1995 | 1983-2002 |
| 18 | Guadiana | Badajoz | 48530.00 | 1995-2002 | - |
| 19 | Guadalquivir | Sevilla | 48548.00 | - | - |
| 20 | Rhine River | Maxau | 50196.00 | 1983-2002 | 1985-1995 |
| 21 | Duero | Fonfria | 63160.00 | 1983-1995 | 1990-1995 |
| 22 | Danube River | Achleiten | 76653.00 | 1983-2002 | 1985-1995 |
| 23 | Ebro, Rio | Tortosa | 84230.00 | 1983-2002 | 1990-1995 |
| 24 | Vistula | Warsaw | 84945.17 | 1983-1990 | 1992-2002 |
| 25 | Rhine River | Mainz | 98206.00 | 1983-2002 | 1985-1995 |
| 26 | Danube River | Korneuburg | 101536.60 | 1996-2002 | - |
| 27 | Rhine River | Kaub | 103488.00 | 1983-2002 | 1985-1995 |
| 28 | Oder River | Hohensaaten-Finow Ap | 109564.00 | 1983-2002 | 1992-2002 |
| 29 | La Loire | Montjean-sur-loire | 109930.00 | 1983-2002 | 1983-1996 |
| 30 | Danube River | Bratislava | 131331.00 | 1983-2002 | 1995-1995 |
| 31 | Rhine River | Lobith | 160800.00 | 1983-2002 | 1983-1995 |
| 32 | Danube River | Nagymaros | 183533.00 | 1983-1999 | 1983-1996 |
| 33 | Vistula | Tczew | 193922.91 | 1983-2002 | 1992-2002 |
| 34 | Danube River | Pancevo | 525009.00 | 1983-2002 | - |
| 35 | Danube River | Ceatal Izmail | 807000.00 | 1983-2002 | - |

**Table A1.** Stations used for the evaluation of the numerical stability and quality of the simulated discharge and stream temperature.