# Peer review of "Hydrological modelling on atmospheric grids; using graphs of sub-grid elements to transport energy and water"

_EGUsphere, 2022_

## Referee Comment (RC1)

**Geoscientific Model Development - Discussions**
**egusphere-2022-690**

**Review: "Hydrological modelling on atmospheric grids; using graphs of sub-grid elements to transport energy and water"**

**J. Polcher, A. Schrapffer, E. Dupont, L. Rinchiuso, X. Zhou, O. Boucher, E. Mouche, C. Ottlé, J. Servonnat (2022)**

**General comments**

The authors present and evaluate a methodology that divides grid cells of a land surface model (LSM) into hydrological transfer units (HTUs). This method allows for a finer and more realistic representation of the river discharge and the energy transport (stream temperature) than a calculation of these parameters directly on the coarser LSM grid. The authors show that their method is independent of the original LSM grid and the hydrological digital elevation model used to construct the HTUs.

This approach is very interesting, as it allows for a better representation, at higher resolution, of hydrological extreme events, but it also enables an integration of the influence of human infrastructure and water usage in Earth System Models. However, it is sometimes difficult to follow the explanations of the authors and to relate them to the results shown on the figures. Therefore, I would invite the authors to review in depth their manuscript. I hope that my comments listed below can be helpful.

I have four major comments:

1) The whole text should be carefully reread as there are many errors (see the "technical corrections" below for some examples), such as missing third person "s", wrong conjugation, etc. The authors are also very parsimonious about the usage of the comma, making some longer or more complex sentences difficult to understand by the reader.

2) The explanations might need to be reformulated or sometimes even restructured, especially in the more technical parts, like section 3, to make it easier to follow for the reader. Some examples are listed in the specific comments below.

3) The authors often use the word *grid*, when they actually mean a single *grid cell* (or *grid point*, or even *grid mesh* or *grid box*), which leads to some confusion for the reader. Especially in the description of the method (section 3), it makes the understanding of the explanations really difficult. This has to be carefully corrected by the authors through the whole manuscript, ideally by using consistently one single denomination. Some examples are:

l. 58: "from grid *cell* to grid *cell*"

l. 179: "atmospheric grid": Is it the entire grid or one grid cell?

l. 186: "arrows pointing out of the grid *cell*" ?

l. 202: "less than 10% of the grid *cell* area" ?

Figure 2: "atmospheric grid *cell*", 2 times ?

l. 229: "atmospheric grid *cell*"?

l. 230: "neighbouring grid *cell*"?

l. 232: "flow out of the *mesh*"

l. 234: "atmospheric grid *cell*"?

l. 236: "*grid box*"

l. 440: "per atmospheric grid *cell*"?

4) Section 5
- Concerning the title: as I understand it, this section is more a sensitivity analysis on some HTU and atmospheric grid parameters. I would expect something else under "numerical implementation" (e.g., code performance, scaling tests, etc.).

- To my opinion, this section should be reorganised and better justified. Especially, the explanations why and how this analysis is performed, need to be clarified. For example, the explanation that forcing data at coarser temporal and spatial resolution are used, because high-resolution data are not available yet, might be moved from the end (subsection 5.4) to the introduction of this section. The authors might also add a short discussion on how a sensitivity analysis on the parameters tested here is influenced (or not) by the low-resolution forcing data. One could think that the temporally (from daily to hourly) and spatially interpolated forcing data (l. 384), and the resulting smoothed discharge (e.g., no sub-daily discharge peaks, an underestimated spatial heterogeneity) weakens the analysis presented here. Some examples are:

l. 416-417: Can conclusions from a simulation forced with interpolated daily data be drawn on peak discharges?

l. 430-433: Thus, is this comparison really relevant? The aim is to reach high-resolution at "low costs" for the routing. So, the comparison should be done with high-resolution data to be more robust.

Section 5.2: Are the very good results shown here not due to the interpolation of coarse forcing data? Would a high-resolution forcing to evaluate the information loss when using less HTUs not be more relevant here? From what I understand from this analysis, i.e., that there is almost no performance gain/loss when changing nbmax, I would chose a low nbmax, or even no HTUs at all (to avoid the issue mentioned in l. 446). Thus, the authors might want to clarify this analysis and the conclusions one might deduce from it.

l. 443: In l. 449, it is stated that all stations considered here have a large up-stream area, thus only large catchments are analysed here. So how can the authors conclude that the results do not depend on the catchment size in l. 443?

l. 464: Could the difference not also have a stronger influence on small catchments? Could it not even be (partly) balanced out over large catchments?

Section 5.4: I understand this subsection as a kind of conclusion of section 5, which is useful. However, the authors mainly focus on the time step here, while other parameters were discussed before, too. If this subsection is meant to focus on the time step, it might be more relevant to move it to subsection 5.1.

l. 470-471: The gx parameters are determined for HDEMs, not for different grids, thus the statement saying that they do not need to be adjusted to the atmospheric grid might be true, but it has not been tested here.

**Specific comments**

l. 22: I would suggest to replace *"Thus"* by *"For Example"*, as this sounds more like an example than a general deduction of the previous sentence.

l. 33-35: While I agree that lateral water movements require a high resolution to be represented in a realistic way, I would say that this is also true for the atmosphere, depending on which processes are of interest. One might think about modelling urban canyons, for example. The authors might clarify that the atmosphere does not need such a high resolution to properly resolve the processes regional atmospheric modelling / land surface modelling usually focuses on.

l. 50: The authors announce two approaches in l. 37. Then, after having introduced these two approaches, they continue with "A complementary methodology…" in l. 50. Would this then be a third approach?

l. 59: "the two linked to the grid": I do not understand what is linked to the grid.

l. 65: Schrapffer et al. (2022) is not listed in the bibliography.

l. 95 equation (1) (and others): What do "j" and "W" stand for?

l. 155-156: This sentence might need to be rewritten.

l. 162: This is also true for the Antarctic region, isn't it? Maybe rephrase to "which do not cover the polar regions"?

l. 169: "with the coarser *atmospheric* mesh" to make it more clear?

l. 181: "The example *over a part of the Rhone valley* in Figure 2c) (*nbmax, which is set to 18 here, will be discussed further below*)" might be clearer.

l. 184-185: This is not clear to me. I understand that the authors still base their explanation on Figure 2c), where there are many HTUs (colours) for the outlet in the SW corner, and not only one as stated here.

l. 189-193: I do not see where the authors consider the two types of confluences presented here in the explanation below (from l. 194 on). Further, it is not clear to me how these two types are differentiated (on the basis of a threshold? If yes, which parameter and which value?).

l. 193: At which threshold is the subdivision too small (< 10% ?) ? And why is there still a need to divide the HTU into two parts if the tributary's confluence is moved downstream?

l. 212-214: This explanation is not clear to me.

l. 213: Should the sums not be computed along all streams *down* to the outflow point?

l. 219: What do the authors mean by "surface groundwater"?

l. 226-227: This sentence should be rephrased to make it easier to understand.

l. 229: It is not clear to me whether the authors want to say that the HTUs *flow* or the atmospheric grid cell *flows* into the ocean.

l. 245, 583, 585: Do the author mean *land surface models* instead of *land system models*?

l. 275: What it the total error? Maybe add something like "from the total error *for each HTU as described above*."

l. 294: Where on Fig. 4 do the authors see that the dz of HTUs is smaller than the sub-segment by over 15% ?

l. 305: "the same results for *the grid with* the highest resolution"?

l. 314-328: As HydroSHEDS does not provide a hydrologically corrected topography (see l. 320), does this whole comparison make sense? Are these results really comparable? Is this comparison not more an analysis of the differences between a hydrologically corrected DEM and a not-corrected one? If this is the case, this comparison might be out of the scope of this paper, to my opinion.

l. 315: "and are better than 5% for both the elevation change and length of samples": I do not understand what is better or compared to what they are better. Maybe some words are missing here.

l. 317: The differences when using HydroSHEDS instead of MERIT seem quite large to me, so I would not write that "the behaviour changes slightly".

l. 330: "are analysed for the Danube *as an example*." ?

l. 342: "are quite constant except for short segments": This is not clear to me. Is the reader supposed to be able to come to this conclusion when looking on Figure 5?

l. 344-346: This statement might be rephrased as it can be understood as if the authors determined the optimal truncation on the basis of computational costs, instead of the result of a t-test.

l. 387: "only  *35 stations* were selected" ? Why these stations?

l. 391: Which reference configuration do the authors mean? The WFDEI-GPCC based simulation from l. 383?

l. 396: Why 225s? Where does this value come from?

l. 399: "only the annual mean is shown": What do the authors mean? The average over one year (which one), or the total average over 1983-1993?

l. 408: "close to or lower to the recommended value": It is not clear to me which value the authors refer to.

l. 462: What does "MEDCORDEXHS" stand for?

l. 498-506: I do not understand this analysis, and especially how it relates to and interprets the results shown on Figure 11. For example, l. 502: "Based on the analysis above, we know that if the forcing *is* the same…": It is not clear to me how one comes to this conclusion.

l. 519: I do not agree that the annual cycle is closer to observations for the Danube. As I see it on Fig. 12, for the Rhine the difference varies between -5 and -2K, while for the Danube it varies between -4 and 0K.

l. 547: "by setting *the scaling parameter* a=10^5 *(eq. 10)*". Remembering what "a" stands for might be useful here.

l. 555: "for both runoff and drainage *(WFDEI_Top)*"?

l. 566: Is winter really the low flow period for the Rhine, the Elbe, the Loire, etc.?

l. 645: Which HDEMs are the authors talking about? MERIT and HydroSHEDS are already made available by their authors.

Table 1: WFDEI → (Weedon et al., 2014)

Table 3: The caption does not really describe the content of the table.

Figure 1:
- It might be useful, for example for l. 276, to also show the entire grids, e.g., as insets, as well as the main rivers mentioned in this paper.

- "The green colour *indicates*"

- "over the actual land-sea mask shown in yellow/blue."

Figure 2:
- maybe mark the rivers mentioned in l. 190-191 on Figure 2a?

- limit the scale to 18 colours

- explanation l. 263-267: the blue line does not exactly follow the white arrows. But if I understand it right, the calculation discussed here is based on HDEM data corresponding to the white arrows (see l. 269), thus the blue line should exactly follow them.

- HTU 8 taken as example in l. 269-270 might be coloured/highlighted in a way that makes it easier to identify it on the figure.

- the description in l. 272-274 is difficult to follow on the figure. May it be useful to highlight the elements mentioned here, or maybe to show them on a separate figure?

Figure 3:
- " provides"

- I would strongly recommend to present these results as box-whisker plots. One coloured box-whisker plot for each river and for each truncation. This would be much more meaningful. It would also avoid an overlap of the curves and lines as it is the case in the current version of this figure. Further, it would then certainly be possible to show all five rivers (add Rhine and Elbe) without overloading the figure.

Figure 4:
- It might be helpful to add the meaning of the solid and dashed lines in the caption.

- I would only show one legend for all sub-figures, at it is always the same, and increase the font size, as it is barely readable.

- It might also be useful to only list as X axis labels the nbmax values for which there are results.

- There are many points missing on the lines, e.g., for the Rhone dz at 25 and 55.

Figure 5:
- It might be helpful to add the meaning of the solid and dashed lines in the caption.

- I would only show one legend for all sub-figures, at it is always the same, and increase the font size, as it is barely readable.

- What does "Danube10" in the X axis titles mean?

Figure 6:
- Caption: add the meaning of the black horizontal line and information on the simulation shown here (WFDEI-MERIT, period, etc.).

Figure 7:
- Caption: add the meaning of the black horizontal line

Figure 8:
- Caption: add the meaning of the black horizontal line. In addition, I only see three different grids, not four as mentioned here.

Figure 10:
- The Y axis labels are barely readable.

- It might be useful to add in the caption which HDEMs are used. "another HDEM" does not give any useful information.

Figure 11:
- "comparing the  *simulated* monthly … to observations"?

Figure 12:
- "Mean annual cycle of *monthly mean* river discharge …" ?

Figure 13:
- "stream temperature *is* available"?

**Technical corrections**

l. 35: "The hydrological community *has* been free"

l. 48: "the horizontal atmospheric grid *is* compatible"

l. 61-62: "… as the hydrological information, which cannot … flow, is treated …"

l. 73: "we will show with *the* ORCHIDEE *LSM* that" as it is the first time it is mentioned in the main text.

l. 144: Should it not be "from 20° *West* to 60° *East*" ?

l. 161: (Nguyen-Quang et al., 2018)

l. 179: "is *built*"

l. 190: "as *illustrated*"

l. 195: "If the subdivision (1) or (2) *is* too small"

l. 223: "to select a *value* of nbmax"

l. 225: "on all grid points of the atmospheric *grid*" ?

l. 234: "HTUs which *flow*"

l. 237-238: "This leads to *situations* like HTU 6 in *Figure* 2d)"

l. 242: "an optimal number of *HTUs*"

l. 245: "a precious *tool*"

l. 250: "each *station*"

l. 261: "represent *such a* segment"?

l. 263: "the *selected* truncation"

l. 264: "the outflow points *belong*"

l. 285: "range *of* +- 10%"

l. 293: "*elevation* changes"?

l. 357: "the reservoir content *is* updated."

l. 428: "This *result* is"

l. 430: "it has to *be* kept in mind"

l. 488: "atmospheric *forcings*"

l. 569: "*Diepoldsau*"

l. 590: "which *combines*"

l. 600: "if the water continuity equation *could not* be solved"

l. 610: "The simulated discharge *is* not"

l. 618: "much more *elaborated* schemes"

l. 636: "an extremely *powerful* tool"

l. 654: "XZ *contributed*"

---

## Referee Comment (RC2)

**Review of**

Hydrological modelling on atmospheric grids; using graphs of sub-grid elements to transport energy and water

**by**

J. Solcher, A. Schlaffer, E. Dupont, L. Rinchiuso, Z. Zhoi, O. Boucher, E. Mouche, C. Ottle, J. Servonnat

**Review date:** 28 Nov 2022

The authors propose a decomposition of atmospheric grid into so-called hydrological transfer units (HTUs) to resolve water flow on the surface more accurately. The proposed decomposition is based on a digital elevation models (HDEM) which contain flow directions. Their method introduces a truncation parameter to reduce memory requirements especially when the river flow is sufficiently resolved. The truncation parameter is dependent on the resolution of the atmospheric grid and the HDEM models in use, and is chosen to minimize a so-called topological error of the flow. Furthermore, the authors rightfully argue that the time steps a reasonable time step can be achieved in their approach, which they demonstrate in great details in a series of numerical experiments for several important grids.

This manuscript presents an interesting and a computationally useful numerical approach with a detailed study of parameters and the comparison thereof in several important studies of surface water flows. I suggest to publish this paper after a minor review.

**Remarks:**

L 32 "as not to introduce any discontinuity"
How does this discontinuity arise? Does this refer to the first approach mentioned in the line 38? If yes, could you please refer the discontinuity to the "first approach"?

L 36 "which is ofter kilometric" -> "which is on a km-scale"

L 69 HTU is used but not introduced

L 70 What comes after "In a first step" is a list of things that you will be addressing in the paper. Could you specify where the items in this list are addressed in the paper, and also to make sure that they are really addressed.

L 73 "simplification of the digital elevation models"
It might be a bis misleading to call then models when in fact they are just data sets, or there something more to it?

L 82 "and covers the fraction" -> "and covers the area fraction"

L 87 "Because we are in a directional graph I+1 is unique"
This needs reformulating. Something along the lines of "$HTU_{I+1}$ is unique"

L 87 "at one point should be the ocean or a water body for endorheic basins"
To what point are you referring to?

L 93 I suggest to have a consistent referring to equations in the manuscript. Sometimes it is referred to as "Eq 1", and later in text it is referred to as "equation 1". In addition, $W_{I,stream}$

should not have italic letter for "stream", rather it should be W_{l,\mathrm{stream}}. The same goes for units - they should not be in the italic letters.

L 104 There is no need for brackets around \lambda_{l,stream}

L 106 In Eq (6) having dz in italic is a bad example, \mathrm{dz} is more appropriate.

L 127 There is no need for brackets around variables in the text.

L 145 "The hydrological data sets" -> "The hydrological data sets HDEMs"

L 148 "30arcsec" -> "30 arcsec" and arcsec should not be in italic

L153 "As we will show…"
Could you please make a reference where is this shown?

L 163 "60arcmin" -> "60~\mathrm{arcsec}" if you write in LaTeX

L 166 Maybe rename to "Supermesh between an atmospheric grid and HDEM"

L 169 "the list of polygons of intersecting polygons" -> "the list of intersecting polygons"

L 180 "Their upstream area is computed  according to the HDEM"
It has not been revealed how this is exactly performed or meant to be. Could you elaborate here a bit more?

L 182 A variable nbmax is mentioned without a meaning to it. Could you give us more information on this variable. Also, instead of writing "nbmax" one could conveniently use a shorter N_{\mathrm{max}}.

L 184 "they contribute remains correct"
In what sense "correct"?

L 185 "in a single and same" -> "in a same"

L 185 Please reformulate the sentence starting with "This first step"

L 197 "HTUS" -> "HTUs"

L 237 "connex"
Do you mean convex? Why bringing up this property of HTU?

L 269 The definition of the cellular error is a bit vague. Is it possible to give a more precise definition of the cellular and the total error?

L 348 We learn that g_X is the inverse of velocity. This should have been also mentioned directly after Eq 5 in L 101.

L 362 "of the HTU : the stream" -> "of the HTU: the stream"

L 399 "in figure 6" -> "Fig. 6"

L 400 "x-axis" -> "$x$-axis" in LaTeX

L 430 "But it has to kept" -> "But it has to be kept"

L 435 At the beginning of Sec. 5.2., could you please again mention the benefits of having nbmax as small as possible?

L 440 "y-axis" -> "$y$-axis" in LaTeX

L 547 I have not seen a study on the scaling parameter "a" but here set to 10^5. Is it possible to specify values earlier in experiments?

L 598 "hydrological Transfer Unit (HTU)" -> "hydrological transfer unit (HTU)"

---

## Author Comment (AC3)

**Reply to Review #1**

Dear Colleague,

thank you very much for very detailed review of our paper. Your comments are very constructive and they have allowed us to improve the manuscript significantly.

**General comments**

The authors present and evaluate a methodology that divides grid cells of a land surface model (LSM) into hydrological transfer units (HTUs). This method allows for a finer and more realistic representation of the river discharge and the energy transport (stream temperature) than a calculation of these parameters directly on the coarser LSM grid. The authors show that their method is independent of the original LSM grid and the hydrological digital elevation model used to construct the HTUs.

This approach is very interesting, as it allows for a better representation, at higher resolution, of hydrological extreme events, but it also enables an integration of the influence of human infrastructure and water usage in Earth System Models. However, it is sometimes difficult to follow the explanations of the authors and to relate them to the results shown on the figures. Therefore, I would invite the authors to review in depth their manuscript. I hope that my comments listed below can be helpful.

I have four major comments:

1) The whole text should be carefully reread as there are many errors (see the "technical corrections" below for some examples), such as missing third person "s", wrong conjugation, etc. The authors are also very parsimonious about the usage of the comma, making some longer or more complex sentences difficult to understand by the reader.

This is true and we have made an effort to improve the text to reduce the number of errors and facilitate the reading. Your comments have contributed to this effort.

2) The explanations might need to be reformulated or sometimes even restructured, especially in the more technical parts, like section 3, to make it easier to follow for the reader. Some examples are listed in the specific comments below.

We hope that the revisions we have made to section 3 through your own comments and those of the other two reviewers have helped to make the description of the algorithm clearer.

3) The authors often use the word *grid,* when they actually mean a single *grid cell* (or *grid point,* or even *grid mesh* or *grid box*), which leads to some confusion for the reader. Especially in the description of the method (section 3), it makes the understanding of the explanations really difficult. This has to be carefully corrected by the authors through the whole manuscript, ideally by using consistently one single denomination. Some examples are:
l. 58: "from grid *cell* to grid *cell"*
l. 179: "atmospheric grid": Is it the entire grid or one grid cell?
l. 186: "arrows pointing out of the grid *cell"* ?
l. 202: "less than 10% of the grid *cell* area" ?
Figure 2: "atmospheric grid *cell",* 2 times ?

l. 229: "atmospheric grid *cell*"?
l. 230: "neighbouring grid *cell*"?
l. 232: "flow out of the *mesh*"
l. 234: "atmospheric grid *cell*"?
l. 236: "*grid box*"
l. 440: "per atmospheric grid *cell*"?

Yes, your are correct. We realized the semantic difficulties while working on the project and have adopted the following vocabulary : grid cells for the elements of the atmospheric grid/mesh and pixels for the cells of the HDEM mesh.

As this is a central distinction in the method which needs to be well understood by the reader we have adopted the following definitions throughout the text and not worried about the repetition of words. More cases than those identified by the reviewer were identified.

Furthermore the following sentence has been added to section 3: "To clarify the discussion, we name here the polygons building the atmospheric mesh grid cells and the one of the HDEMs pixels."

4) Section 5
- Concerning the title: as I understand it, this section is more a sensitivity analysis on some HTU and atmospheric grid parameters. I would expect something else under "numerical implementation" (e.g., code performance, scaling tests, etc.).

This section aims to evaluate the stability and convergence of the numerical implementation of the HTU graphs. It should inform the user of these methods of how many subdivisions of the grid cells is appropriate, which is the largest time step which can be used before results degrade and how important the choice of the HDEM is. We agree that "numerical implementation" is indeed too short of a title and we propose to use "Evaluation of numerical implementations".

- To my opinion, this section should be reorganised and better justified. Especially, the explanations why and how this analysis is performed, need to be clarified. For example, the explanation that forcing data at coarser temporal and spatial resolution are used, because high-resolution data are not available yet, might be moved from the end (subsection 5.4) to the introduction of this section. The authors might also add a short discussion on how a sensitivity analysis on the parameters tested here is influenced (or not) by the low-resolution forcing data. One could think that the temporally (from daily to hourly) and spatially interpolated forcing data (l. 384), and the resulting smoothed discharge (e.g., no sub-daily discharge peaks, an underestimated spatial heterogeneity) weakens the analysis presented here.

In this evaluation of the routing method through HTU graphs, no attempt is made to evaluate the discharge at temporal resolution below daily means. This has been clarified in the introduction to section 5 with the last sentence :

"The chosen methodology limits the numerical evaluation of the scheme presented here to the spatial resolutions of the atmospheric features of about 0.5° and daily mean discharge values."

Some examples are:
l. 416-417: Can conclusions from a simulation forced with interpolated daily data be drawn on peak discharges?

No and this is outside of the scope of the current evaluation of the scheme as explained above.

l. 430-433: Thus, is this comparison really relevant? The aim is to reach high-resolution at "low costs" for the routing. So, the comparison should be done with high-resolution data to be more robust.

Section 5.2: Are the very good results shown here not due to the interpolation of coarse forcing data? Would a high-resolution forcing to evaluate the information loss when using less HTUs not be more relevant here? From what I understand from this analysis, i.e., that there is almost no performance gain/loss when changing nbmax, I would chose a low nbmax, or even no HTUs at all (to avoid the issue mentioned in l. 446). Thus, the authors might want to clarify this analysis and the conclusions one might deduce from it.

This is clearly spelled out in the last paragraph of section 5. In order to move to the evaluation of the model to represent sub-diurnal features of the lateral water movements adequate forcing data sets would be needed. This is not only valid for the routing scheme but also ORCHIDEE as a whole. We know that as we will move our land surface model to km-scale resolutions precipitation and other atmospheric drivers will be more variable in space and time. Are our land surface models able to reproduce correctly the impact of this higher variability on the land surface processes ? This is an open question today in the scientific community especially with the Digital Earths effort underway.

As we are not there yet, we believe that it is perfectly justifiable to validate the routing network with the current state of the art atmospheric forcing data.

l. 443: In l. 449, it is stated that all stations considered here have a large up-stream area, thus only large catchments are analysed here. So how can the authors conclude that the results do not depend on the catchment size in l. 443?

As explained in the last paragraph of the introduction to section 5, the validation is limited to catchments with areas from about $10^3$ to $10^6$ km$^2$. The lower limit is given by the grid area on which the atmospheric forcing is available and the upper limit by the Danube. No claims are maid for smaller or larger catchments.

l. 464: Could the difference not also have a stronger influence on small catchments? Could it not even be (partly) balanced out over large catchments?

Yes, there is naturally a compensation of errors from smaller catchments when they converge with larger flow. A simple dilution process occurs here. This is why it is important to evaluate any routing scheme over a wide range of catchment areas as done here.

Section 5.4: I understand this subsection as a kind of conclusion of section 5, which is useful. However, the authors mainly focus on the time step here, while other parameters were discussed before, too. If this subsection is meant to focus on the time step, it might be more relevant to move it to subsection 5.1.

Yes, solved by expanding first sentence :

"It is an important result that for the range of atmospheric grids tested here, and optimal truncation and time step can be selected according to the criteria defined above which provides a converged solution. That is, the simulated stream flow and temperatures are relatively insensitive to higher truncation or shorter time steps thus optimizing the numerical cost of the model."

l. 470-471: The gx parameters are determined for HDEMs, not for different grids, thus the statement saying that they do not need to be adjusted to the atmospheric grid might be true, but it has not been tested here.

Yes, it is an hypothesis and expressed as such in the paper. The explanation follows in the first paragraph of section 5. We indeed hypothesize that the partition of water leaving the unsaturated zone as runoff or drainage is probably a more important factor for the choice of $g_X$.

An option to test this would be to change the infiltration parameterization of ORCHIDEE and evaluate the impact on river discharge. Can then $g_X$ be adjusted in order to obtain the same results again ? In other words, can the change of residence time of water in the unsaturated zone be compensated by the residence time in the saturated soils of the model ?

These tests should be performed once the representation of the saturated zone within ORCHIDEE is on a strong physical footing. Else errors or adjustments in one part of the water cycle are compensated by another component.

**Specific comments**

l. 22: I would suggest to replace *"Thus"* by *"For Example",* as this sounds more like an example than a general deduction of the previous sentence.

Corrected

l. 33-35: While I agree that lateral water movements require a high resolution to be represented in a realistic way, I would say that this is also true for the atmosphere, depending on which processes are of interest. One might think about modelling urban canyons, for example. The authors might clarify that the atmosphere does not need such a high resolution to properly resolve the processes regional atmospheric modelling / land surface modelling usually focuses on.

If we represent urban canyons in atmospheric models, then we would also need to simulate the gullies which carry rainwater back to their natural flows. I believe it lies in the fact that liquid water and its flows is more heterogeneous  (gathered by gravitation ?) than water vapor.

l. 50: The authors announce two approaches in l. 37. Then, after having introduced these two approaches, they continue with "A complementary methodology…" in l. 50. Would this then be a third approach?

We prefer to label it as a complementary method to the atmospheric grid cell to grid cell method. It only refines it by introducing a hydrological tilling.

l. 59: "the two linked to the grid": I do not understand what is linked to the grid.

This is rephrased as : "In the list of criteria established by (Kauffeldt et al. 2016) to classify large-scale hydrological models, a hybrid routing addresses in particular the two linked to the grid (Flexibility to grid structure and to grid resolution)."

l. 65: Schrapffer et al. (2022) is not listed in the bibliography.

That is corrected now.

l. 95 equation (1) (and others): What do "j" and "W" stand for?

Corrected

l. 155-156: This sentence might need to be rewritten.

We do not see the issue with this sentence.

l. 162: This is also true for the Antarctic region, isn't it? Maybe rephrase to "which do not cover the

polar regions"?

Done

l. 169: "with the coarser *atmospheric* mesh" to make it more clear?

Done

l. 181: "The example *over a part of the Rhone valley* in Figure 2c) (*nbmax, which is set to 18 here, will be discussed further below*)" might be clearer.

Corrected

l. 184-185: This is not clear to me. I understand that the authors still base their explanation on Figure 2c), where there are many HTUs (colours) for the outlet in the SW corner, and not only one as stated here.

Figure 2 does not contain a panel for the first phase of the construction of HTUs. So what is seen in figure 2c) is the result of the phase described in 3.6.

l. 189-193: I do not see where the authors consider the two types of confluences presented here in the explanation below (from l. 194 on). Further, it is not clear to me how these two types are differentiated (on the basis of a threshold? If yes, which parameter and which value?).

The distinction is performed with the global upstream area which is first ordered. So it can be determined which is the largest tributary which needs to be sub-divided.

l. 193: At which threshold is the subdivision too small (< 10% ?) ? And why is there still a need to divide the HTU into two parts if the tributary's confluence is moved downstream?

The threshold for the sub-division is set to grid_cell_area/nbasmax. This allows the parameter to adapt to various truncation.

l. 212-214: This explanation is not clear to me.

It is just that the parameters are computed to all rivers within the HTU and then averaged over all these rivers to provide a HTU-mean value.

l. 213: Should the sums not be computed along all streams *down* to the outflow point?

Yes, "down to" is clearer than "up to" in this case. The text has been changed to clarify !

l. 219: What do the authors mean by "surface groundwater"?

"near surface groundwater". Groundwater exists at various horizons and only the one closest to the surface is considered here.

l. 226-227: This sentence should be rephrased to make it easier to understand.

Reformulated to "The merger of HTUs will be performed by always favoring the largest HTU or the one with the largest upstream area while trying to preserve the diversity of outflow directions out of the atmospheric grid cell."

l. 229: It is not clear to me whether the authors want to say that the HTUs *flow* or the atmospheric grid cell *flows* into the ocean.

Corrected

l. 245, 583, 585: Do the author mean *land surface models* instead of *land system models*?

We believe both can be used.

l. 275: What it the total error? Maybe add something like "from the total error *for each HTU as described above*."

In this paragraph we have revised the definition of the cell error. Perhaps it will also clarify what we mean by total error on the river segment.

l. 294: Where on Fig. 4 do the authors see that the dz of HTUs is smaller than the sub-segment by over 15% ?

The errors plotted in Figure 4 are those obtained by comparing the properties of the river segment (length or elevation change) computed on the HDEM and those computed on the graph of HTUs. The full blue line is at -10% indicating that the elevation change in the HTU graph is lower than in the HDEM.

l. 305: "the same results for *the grid with* the highest resolution"?

Corrected

l. 314-328: As HydroSHEDS does not provide a hydrologically corrected topography (see l. 320), does this whole comparison make sense? Are these results really comparable? Is this comparison not more an analysis of the differences between a hydrologically corrected DEM and a not-corrected one? If this is the case, this comparison might be out of the scope of this paper, to my opinion.

We believe it shows the value of hydrological correction of topography. It is interesting to see that this error in dz does not impact strongly the simulated discharge over the catchments considered (Figure 10).

l. 315: "and are better than 5% for both the elevation change and length of samples": I do notunderstand what is better or compared to what they are better. Maybe some words are missing here.

The sentence was split in two to make it simpler.

l. 317: The differences when using HydroSHEDS instead of MERIT seem quite large to me, so I would not write that "the behaviour changes slightly".

Yes, for dz the differences are large but we know that this is because of the lack of  hydrologically corrected topography in HydroSHEDS. More interesting is the evolution of the error on the length. There indeed both have a convergence of the errors at about nbmax of 25 but the speed at which they reach the minimal errors are different. This is the noteworthy difference.

l. 330: "are analysed for the Danube *as an example*." ?

Corrected

l. 342: "are quite constant except for short segments": This is not clear to me. Is the reader supposed to

be able to come to this conclusion when looking on Figure 5?

This sentence presents one of the results which is not shown as indicated in the previous sentence.

l. 344-346: This statement might be rephrased as it can be understood as if the authors determined the optimal truncation on the basis of computational costs, instead of the result of a t-test.

Thank you. The sentence is clarified.

l. 387: "only a few *35 stations* were selected" ? Why these stations?

Figures 6 to 10 would be unreadable if more than 35 stations would have been used. The choice has only been driven by the sampling of catchment sizes and climates covered.

l. 391: Which reference configuration do the authors mean? The WFDEI-GPCC based simulation from l. 383?

No, for each tested parameter a reference configuration has been selected (the one expected to have the smallest numerical error) and all other simulations were compared to it. For each test (section 5.1, 5.2 and 5.3) the reference configuration is explained.

It is the smallest value proposed by the criteria proposed in section 5 to fulfill the CFL conditions in 75% of all HTUs applied to all graphs generated.

l. 399: "only the annual mean is shown": What do the authors mean? The average over one year (which one), or the total average over 1983-1993?

Corrected to "only the mean is shown". The mean over the entire period is considered here.

l. 408: "close to or lower to the recommended value": It is not clear to me which value the authors refer to.

The recommended value is the one produced by the method described between lines 361 and 378.

The MEDCORDEX grid with the HTU graph computed using HydroSHEDS. The acronym has been added to the caption of table 3.

l. 498-506: I do not understand this analysis, and especially how it relates to and interprets the results shown on Figure 11. For example, l. 502: "Based on the analysis above, we know that if the forcing *is* the same…": It is not clear to me how one comes to this conclusion.

This follows directly from figure 10 where correlation and ratio of standard deviation for simulated discharges at the 35 stations are displayed. These metrics can be transferred directly to the Taylor diagrams of figure 11. It shows that the atmospheric grid or HTU graph selected cannot explain the difference found between both forcings.

l. 519: I do not agree that the annual cycle is closer to observations for the Danube. As I see it on Fig.

12, for the Rhine the difference varies between -5 and -2K, while for the Danube it varies between -4 and 0K.

Yes, but on the Danube the model does not have a general bias like on the Rhine. We have clarified the sentence by adding "… in particular in summer".

l. 547: "by setting *the scaling parameter* a=10^5 *(eq. 10)*". Remembering what "a" stands for might be useful here.

It is just a relaxation constant which is explained in lines 128-135.

l. 555: "for both runoff and drainage *(WFDEI_Top)*"?

Added !

l. 566: Is winter really the low flow period for the Rhine, the Elbe, the Loire, etc.?

An incorrect generalization. Corrected to "winter is the period when the flows are dominated by the groundwater contribution in mountain catchments". The discharge plotted in figure 11 are misleading as these are far downstream for both rivers. For smaller catchments, especially those in the mountains, it is clear that winter is the low flow period.

l. 645: Which HDEMs are the authors talking about? MERIT and HydroSHEDS are already made available by their authors.

Both are available on Zenodo.

Table 1: WFDEI → (Weedon et al., 2014)

Corrected

Table 3: The caption does not really describe the content of the table.

Corrected

Figure 1:
- It might be useful, for example for l. 276, to also show the entire grids, e.g., as insets, as well as the main rivers mentioned in this paper.
- "The green colour *indicates*"
- "over the actual land-sea mask shown in yellow/blue."

Corrected

Figure 2:
- maybe mark the rivers mentioned in l. 190-191 on Figure 2a?

It is really small as can be seen in the flow accumulation.

- limit the scale to 18 colours

It is difficult to find a color scale which is differentiated enough.

- explanation l. 263-267: the blue line does not exactly follow the white arrows. But if I understand it right, the calculation discussed here is based on HDEM data corresponding to the white arrows (see l. 269), thus the blue line should exactly follow them.

No the blue line is a straight line between the outflow point of the previous HTU and the outflow of the current. It symbolizes that only the average properties of the rivers are considered.

- HTU 8 taken as example in l. 269-270 might be coloured/highlighted in a way that makes it easier to identify it on the figure.
- the description in l. 272-274 is difficult to follow on the figure. May it be useful to highlight the elements mentioned here, or maybe to show them on a separate figure?

A possibility would be to make out of figure 2 a step by step description of the HTU construction algorithm. But at this stage this seems too complex as the extraction of all the information from the code would need to be set-up so that it can be plotted afterwards.

Figure 3:
- "Figure provides"
- I would strongly recommend to present these results as box-whisker plots. One coloured box-whisker plot for each river and for each truncation. This would be much more meaningful. It would also avoid an overlap of the curves and lines as it is the case in the current version of this figure. Further, it would then certainly be possible to show all five rivers (add Rhine and Elbe) without overloading the figure.

A box-whisker representation would not get around the issue that for each value in the x-direction the information would overlay. So it will remain difficult to read.

Figure 4:
- It might be helpful to add the meaning of the solid and dashed lines in the caption.
- I would only show one legend for all sub-figures, at it is always the same, and increase the font size, as it is barely readable.
- It might also be useful to only list as X axis labels the nbmax values for which there are results.
- There are many points missing on the lines, e.g., for the Rhone dz at 25 and 55.

The caption was improved. When points are missing it means that the change of the error is not significant compare to the previous coarser truncation.

Figure 5:
- It might be helpful to add the meaning of the solid and dashed lines in the caption.
- I would only show one legend for all sub-figures, at it is always the same, and increase the font size, as it is barely readable.
- What does "Danube10" in the X axis titles mean?

Caption improved.

Figure 6:
- Caption: add the meaning of the black horizontal line and information on the simulation shown here (WFDEI-MERIT, period, etc.).

Caption improved.

Figure 7:

- Caption: add the meaning of the black horizontal line

Added.

Figure 8:
- Caption: add the meaning of the black horizontal line. In addition, I only see three different grids, not four as mentioned here.

Corrected.

Figure 10:
- The Y axis labels are barely readable.
- It might be useful to add in the caption which HDEMs are used. "another HDEM" does not give any useful information.

Caption improved.

Figure 11:
- "comparing the observed *simulated* monthly … to observations"?

Caption corrected.

Figure 12:
- "Mean annual cycle of *monthly mean* river discharge …" ?

Caption corrected.

Figure 13:
- "stream temperature *is* available"?

Caption corrected.

**Technical corrections**

l. 35: "The hydrological community *has* been free"
l. 48: "the horizontal atmospheric grid *is* compatible"
l. 61-62: "… as the hydrological information, which cannot … flow, is treated …"
l. 73: "we will show with *the* ORCHIDEE *LSM* that" as it is the first time it is mentioned in the main text.
l. 144: Should it not be "from 20° *West* to 60° *East*" ?

l. 161: (Nguyen-Quang et al., 2018)
l. 179: "is *built*"
l. 190: "as *illustrated*"
l. 195: "If the subdivision (1) or (2) *is* too small"
l. 223: "to select a *value* of nbmax"
l. 225: "on all grid points of the atmospheric *grid*" ?
l. 234: "HTUs which *flow*"
l. 237-238: "This leads to *situations* like HTU 6 in *Figure* 2d)"
l. 242: "an optimal number of *HTUs*"
l. 245: "a precious *tool*"
l. 250: "each *station*"

l. 261: "represent *such a* segment"?
l. 263: "the *selected* truncation"
l. 264: "the outflow points *belong*"
l. 285: "range *of* +- 10%"
l. 293: "*elevation* changes"?
l. 357: "the reservoir content *is* updated."
l. 428: "This *result* is"
l. 430: "it has to *be* kept in mind"
l. 488: "atmospheric *forcings*"
l. 569: "*Diepoldsau*"
l. 590: "which *combines*"
l. 600: "if the water continuity equation *could not* be solved"
l. 610: "The simulated discharge *is* not"
l. 618: "much more *elaborated* schemes"
l. 636: "an extremely *powerful* tool"
l. 654: "XZ *contributed*"

Thank you very much for all these corrections. They were all applied to the manuscript.

---

## Author Comment (AC4)

**Reply to Review #2**

Dear Colleague,

thank you very much for taking the time to review our paper and proposing some very constructive comments. They have contributed to improve the manuscript.

The authors propose a decomposition of atmospheric grid into so-called hydrological transfer units (HTUs) to resolve water flow on the surface more accurately. The proposed decomposition is based on a digital elevation models (HDEM) which contain flow directions. Their method introduces a truncation parameter to reduce memory requirements especially when the river flow is sufficiently resolved. The truncation parameter is dependent on the resolution of the atmospheric grid and the HDEM models in use, and is chosen to minimize a so-called topological error of the flow. Furthermore, the authors rightfully argue that the time steps a reasonable time step can be achieved in their approach, which they demonstrate in great details in a series of numerical experiments for several important grids.

This manuscript presents an interesting and a computationally useful numerical approach with a detailed study of parameters and the comparison thereof in several important studies of surface water flows. I suggest to publish this paper after a minor review.

Thank you very much for these positive general comments on the method presented in the paper.

**Remarks:**

L 32 "as not to introduce any discontinuity"
How does this discontinuity arise? Does this refer to the first approach mentioned in the line 38? If yes, could you please refer the discontinuity to the "first approach"?

The sentence is indeed not very clear. It has thus been reformulated as follows : "in order to avoid the discontinuity which would have been introduced if a finer mesh would have been used for the land surface".

L 36 "which is ofter kilometric" -> "which is on a km-scale"

changed

L 69 HTU is used but not introduced

yes, in line 57 with the sentence : "The combination of both yields graphs of hydrological transfer units (HTU) ..."

L 70 What comes after "In a first step" is a list of things that you will be addressing in the paper. Could you specify where the items in this list are addressed in the paper, and also to make sure that they are really addressed.

Corrected

L 73 "simplification of the digital elevation models"
It might be a bis misleading to call then models when in fact they are just data sets, or there something more to it?

I agree that this denomination of high resolution orographic data is misleading. But it is a standard term used in many disciplines : https://en.wikipedia.org/wiki/Digital_elevation_model

L 82 "and covers the fraction" -> "and covers the area fraction"

Yes, corrected

L 87 "Because we are in a directional graph I+1 is unique"
This needs reformulating. Something along the lines of "HTU_{I+1} is unique"

Yes, corrected : "Because we are in a directional graph the vertex $i+1$ is unique and at some point downstream the graph should end in the ocean or a water body for endorheic basins." should be clearer.

L 87 "at one point should be the ocean or a water body for endorheic basins" To what point are you referring to?

Sentence proposed above should clarify this point as well.

L 93 I suggest to have a consistent referring to equations in the manuscript. Sometimes it is referred to as "Eq 1", and later in text it is referred to as "equation 1". In addition, W_{I,stream} should not have italic letter for "stream", rather it should be W_{I,\mathrm{stream}}. The same goes for units - they should not be in the italic letters.

Systematically the fonts for mathematical symbols are used. So either the equation environment in LaTeX is used order a simple $...$ for in-line mathematical symbols. When the equation is numbered the "Eq x" is used, else "equation" is kept.

L 104 There is no need for brackets around \lambda_{I,stream}

This is to be consistent with the rest of the sentence where "($\lambda_i$)" is used.

L 106 In Eq (6) having dz in italic is a bad example, \mathrm{dz} is more appropriate.

Here again the LaTeX formulation for the in-line mathematical expressions $dz$ is used.

L 127 There is no need for brackets around variables in the text.

Removed

L 145 "The hydrological data sets" -> "The hydrological data sets HDEMs"

The caption of table 2 was changed to "The hydrological digital elevation models (HDEM) used in this study to evaluate the building of routing graph and the simulated river discharge".

L 148 "30arcsec" -> "30 arcsec" and arcsec should not be in italic

Systematically we now write in LaTeX "$30\,arcsec$".

L153 "As we will show…"
Could you please make a reference where is this shown?

Yes, "(section 4)" was added to the sentence.

L 163 "60arcmin" -> "60~\mathrm{arcsec}" if you write in LaTeX

Error corrected : "$60\,arcsec$".

L 166 Maybe rename to "Supermesh between an atmospheric grid and HDEM"

Corrected.

L 169 "the list of polygons of intersecting polygons" -> "the list of intersecting polygons"

Corrected.

L 180 "Their upstream area is computed  according to the HDEM"
It has not been revealed how this is exactly performed or meant to be. Could you elaborate here a bit more?

This sentence has been updated to "Their local upstream area is computed using the the area of the overlapping HDEM pixels.". It is important to know that at this stage only the upstream area local to the atmospheric grid cell can be evaluated.

L 182 A variable nbmax is mentioned without a meaning to it. Could you give us more information on this variable. Also, instead of writing "nbmax" one could conveniently use a shorter N_{\mathrm{max}}.

The sentence was clarified with : for the user selected truncation $nbmax=18$. We prefer to keep "nbmax" to ensure consistency with the graphics.

L 184 "they contribute remains correct" In
what sense "correct"?

The catchment area is preserved.

L 185 "in a single and same" -> "in a same"

Corrected.

L 185 Please reformulate the sentence starting with "This first step"

Reformulated to : "This first step will conclude with as many HTUs as there are arrows pointing out of the grid cell, as illustrated in Figure 2".

L 197 "HTUS" -> "HTUs"

Corrected.

L 237 "connex"
Do you mean convex? Why bringing up this property of HTU?

Connex in the sense of "connected".

L 269 The definition of the cellular error is a bit vague. Is it possible to give a more precise definition of the cellular and the total error?

We have attempted to clarify this with the following addition : "Within each HTU we can compare the sub-segment's properties computed with the HDEM to the one used for the HTU."

L 348 We learn that g_X is the inverse of velocity. This should have been also mentioned directly after Eq 5 in L 101.

This has been added in the presentation of Eq 5.

L 362 "of the HTU : the stream" -> "of the HTU: the stream"

Corrected.

L 399 "in figure 6" -> "Fig. 6"

We now use systematically "Figure n".

L 400 "x-axis" -> "$x$-axis" in LaTeX

There is no need to use the mathematical fonts here in our opinion.

L 430 "But it has to kept" -> "But it has to be kept"

Corrected.

L 435 At the beginning of Sec. 5.2., could you please again mention the benefits of having nbmax as small as possible?

Excellent idea. We added : "Using small values for *nbmax* reduces the memory footprint and computational time of the routing scheme."

L 440 "y-axis" -> "$y$-axis" in LaTeX

As above.

L 547 I have not seen a study on the scaling parameter "a" but here set to 10^5. Is it possible to specify values earlier in experiments?

The relaxation to the surface temperature using the "a" parameter is only a rapid solution to an explicit representation of the energy balance on the open water fraction with the atmospheric grid. It is explained in lines 128-135. This simple parameterization will be replaced by the lake model introduced in the paper of A. Bernus et al. 2022.

L 598 "hydrological Transfer Unit (HTU)" -> "hydrological transfer unit (HTU)"

Corrected.

---

## Author Comment (AC5)

**Reply to Review #3**

Dear Colleague,

thank you very much for taking the time to review our paper and proposing some interesting comments. They have helped us greatly to improve the manuscript.

In this manuscript, the authors proposed a new tiling method to efficiently incorporate high-resolution topographic information for better hydrological simulations by atmospheric models while keeping the atmospheric grids. First, they built hydrological coherent units (HTUs) from hydrological digital elevation models. Then, by evaluating the generated river networks and sensitivity experiment results, the authors proposed a way to find appropriate truncation numbers for HTUs and time steps. Finally, they conducted offline ORCHIDEE simulations and compared the simulated discharge and river temperature with observations.

The manuscript is well-written and contains valuable information for ESM modelers. Therefore, I recommend publication after minor modifications.

Thank you for these encouraging remarks.

General comments:

1. Can you explain more about the connectivity between atmospheric grids? For example, how do you maintain consistency when an HDEM grid overlaps with multiple atmospheric grids and is split into multiple supermeshes?

Yes, this is indeed a complexity which was not discussed in the presentation of the methodology. The following sentences were added to explain the problem and the solution adopted :

"A further complexity is introduced by HDEM pixels overlapping more than one atmospheric cell. All polygons generated by the supermeshing are kept but the largest is considered to dominate and to determine the connectivity of the graph between grid cells."

2. It would be helpful to add a figure to compare the results with the previous ORCHIDEE simulations explained in Section 6. Have you observed any improvements by taking into account the detailed topography information?

The actual routing of ORCHIDEE has not changed. Only the methodology for the construction of the HTU graphs was externalized and extended so that higher resolution HDEMs can be used. It was verified that the current method applied to the Fekete et al. (2000) produces the same result as the previous version of ORCHIDEE (Results in the thesis manuscript of A. Schrapffer 2022) . Running the tests of section 6 also with the Fekete et al. (2000) HDEM makes little sense as it is at

0.5° resolution. Thus is has insufficient resolution to produce enough meaningful HTUs on the grids used by the WFDEI and E2OFD forcing.

L97: Can the atmospheric grid be divided into vegetation tiles? If so, how can vegetation tiles be related to HTUs?

Yes, this is planned future development. The vegetation would have to interact with the superficial groundwater and this can be done as we know the shape of the HTU. An example for the implementation of these ideas can be found in Picourlat et al. 2022 (see references).

L107: It may be helpful to add an explanation of how the lambda was derived.

It is a simplified version of the Manning formula for the average velocity in an open channel.

L182: Eastern -> Western? Could you include a compass symbol to indicate the direction in Figure 2?

Corrected

L185: Are any panels showing the results of the first step in Figure 2?

No, it would chose just to present the final result for two truncations. The intermediate results have not been archived.

L193: It is unclear to me which location you are referring to; it would be helpful if Figure 2a includes the names of the local rivers.

We fear that this will make the graphics too complex if all the small rivers have to be added to figure 2a).

Figure 8: four different grids -> three different grids?

Corrected.

---

## Author Response (AR2)

Dear Sir,

Thank you very much for accepting our manuscript referenced EGUSPHERE-2022-690. Please find below our response to the changes you and the reviewer have asked for.

- Zenodo : Indeed the description of the content were not yet added. We now provide a description of the content of the archive with the following text which should clarify that all the data is there to guarantee reproducibility of the results presented :
  *"The archived file includes the following items in support of the publication in GMD :*
  - *The code of the Routing pre-processor (RoutingPreProc) written in Python and FORTRAN.*
  - *The NetCDF files containing the descriptions of the 4 grids on which the working of RoutingPreProc is exemplified.*
  - *The two high resolution hydrological digital elevation models used in the validation : MERIT and HydroSHEDS. Both are NetCDF files."*
- Figure 2 : We have a technical problem with this figure. The latest version of the code was lost after a disc of Anthony Schrapffer crashed. He could recover an older version of the code. It would take another week or so in order to reconstruct the figure in its current state and then modify the scale. As the proposed deadline is quite short, we though better to leave the figure as it is.
- Figure 3 : As proposed by the reviewer we used "grouped whisker boxplots" to illustrate the statistical distribution of the river segment properties. We must acknowledge that it demonstrates much better the point made in the text with this illustration.
- The technical issues were corrected as suggested.

Best regards

Jan Polcher

---

## Author Response (AR3)

Dear Sir,

Thank you very much for accepting our manuscript referenced EGUSPHERE-2022-690. Please find below our response to the changes you have asked for.

The version on Zenodo has now been revised and more information added. In detail the content of the package with the DOI is now :

**Grids**

The 4 grids for which the constructions and quality of the routing graph were evaluated are provided.

**Data**

This directory contains the hydrological digital elevation models (HDEM) used in the paper to produce the routing graph which are evaluated.

**Documentation**

A copy of the WiKi is provided so that the software can be used with the above data.

- Access to the documentation.

- Evolving documentation on gitlab.

**Routingpp**

The main code as used for the paper. In this directory some standard test cases are also provided.

**Diagnostic Tools**

The diagnostic tools used to analyse the produced routing graph are provided :

- MainStream : The codes to generate and analyse the river segments to evaluate the quality of the graphs.
- validation : Validation of river discharge and temperature simulated by ORCHIDEE using the graphs build with the Routingpp code.

The new Zenodo repository is : https://doi.org/10.5281/zenodo.7788209
This information has been updated in the manuscript but is not well highlighted by latexdiff tool.

We hope that this satisfies the reproducibility ambitions GMD has set for the results to be published in the journal.

Best regards

Jan Polcher